# FluidLab: A Differentiable Environment for Benchmarking Complex Fluid Manipulation

**Zhou Xian**
CMU
zhouxian@cmu.edu

**Bo Zhu**
Dartmouth College
bo.zhu@dartmouth.edu

**Zhenjia Xu**
Columbia University
xuzhenjia@cs.columbia.edu

**Hsiao-Yu Tung**
MIT
hytung@mit.edu

**Antonio Torralba**
MIT
torralba@mit.edu

**Katerina Fragkiadaki**
CMU
katef@cs.cmu.edu

**Chuang Gan**
MIT-IBM Watson AI Lab
ganchuang@csail.mit.edu

## ABSTRACT

Humans manipulate various kinds of fluids in their everyday life: creating latte art, scooping floating objects from water, rolling an ice cream cone, etc. Using robots to augment or replace human labors in these daily settings remain as a challenging task due to the multifaceted complexities of fluids. Previous research in robotic fluid manipulation mostly consider fluids governed by an ideal, Newtonian model in simple task settings (*e.g.*, pouring water into a container). However, the vast majority of real-world fluid systems manifest their complexities in terms of the fluid's complex material behaviors (*e.g.*, elastoplastic deformation) and multi-component interactions (*e.g.* coffee and frothed milk when making latte art), both of which were well beyond the scope of the current literature. To evaluate robot learning algorithms on understanding and interacting with such complex fluid systems, a comprehensive virtual platform with versatile simulation capabilities and well-established tasks is needed. In this work, we introduce *FluidLab*, a simulation environment with a diverse set of manipulation tasks involving complex fluid dynamics. These tasks address interactions between solid and fluid as well as among multiple fluids. At the heart of our platform is a fully differentiable physics simulator, *FluidEngine*, providing GPU-accelerated simulations and gradient calculations for various material types and their couplings, extending the scope of the existing differentiable simulation engines. We identify several challenges for fluid manipulation learning by evaluating a set of reinforcement learning and trajectory optimization methods on our platform. To address these challenges, we propose several domain-specific optimization schemes coupled with differentiable physics, which are empirically shown to be effective in tackling optimization problems featured by fluid system's non-convex and non-smooth properties. Furthermore, we demonstrate reasonable sim-to-real transfer by deploying optimized trajectories in real-world settings. FluidLab is publicly available at: https://fluidlab2023.github.io.

## 1 INTRODUCTION

Imagine you are fishing on the lakeside. Your hat falls into the water and starts to float out of reach. In order to get it back, you use your hands to paddle the water gently, generating a current that pulls the the hat back into reach. In this scenario, the water functions as a transmitting medium for momentum exchange to accomplish the manipulation task. In fact, from creating latte art to making ice creams, humans frequently interact with different types of fluids everyday (Figure 1). However, performing these tasks are still beyond the reach of today's robotic systems due to three major challenges. First, fluid systems exhibit a wide range of material models (e.g., air, liquid,

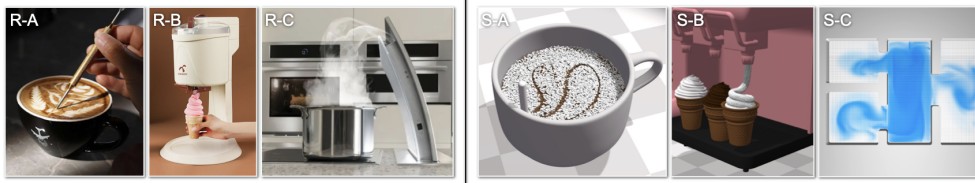

Figure 1: We encounter various manipulation tasks involving fluids in everyday life. Left: real-world scenarios including making latte art, making ice creams and designing air ventilation systems. Right: simulation tasks in FluidLab involving latte art, ice-creams and indoor air circulation.

granular flow, etc. (Bridson, 2015)) that are difficult to simulate within a unified framework. Second, describing the state of a fluid system under manipulation requires a high dimensional state space (Lin et al., 2020). Third, the nonlinear dynamics of different coupling models pose unique challenges to efficient solution search in the state space (Schenck & Fox, 2018). For example, although water and smoke face the same computational challenges in solving fluid equations, manipulating these two mediums in different physical contexts requires different strategies.

Prior works in robotic manipulation covering fluids mostly adopt relatively simple task settings, and usually consider tasks with a single-phase fluid, such as pouring water (Schenck & Fox, 2018; Lin et al., 2020; Sermanet et al., 2018) or scooping objects from water (Seita et al., 2022; Antonova et al., 2022). In this work, we aim to study the problem of *complex fluid manipulation* with more challenging task settings and complex goal configurations.

Since virtual environments have played an instrumental role in benchmarking and developing robot learning algorithms (Brockman et al., 2016; Kolve et al., 2017; Xiang et al., 2020), it is desirable to have a virtual platform providing a versatile set of fluid manipulation tasks. In this work, we propose *FluidLab*, a new simulation environment to pave the way for robotic learning of complex fluid manipulation skills that could be useful for real-world applications. FluidLab provides a new set of manipulation tasks that are inspired by scenarios encountered in our daily life, and consider both interactions within multiple fluids (e.g., coffee and frothed milk), as well as the coupling between fluids and solids. Building such an environment poses a major challenge to the simulation capability: fluids encountered in our life consist of a wide spectrum of materials with different properties; in addition, simulating these tasks also requires modeling interactions with other non-fluid materials. For example, coffee is a type of (mostly inviscid) *Newtonian* liquid; frothed milk behaves as *viscous Newtonian* liquid (i.e., with a constant viscosity that does not change with stress) (Figure 1, R-A), while ice cream is a type of *non-Newtonian* fluid (in particular, a shear-thinning fluid with its viscosity decreasing as the local stress increasing) (Figure 1, R-B); when stirring a cup of coffee, we use a *rigid* stick to manipulate the coffee (Figure 1, R-A). However, there still lacks a simulation engine in the research community that supports such a wide variety of materials and couplings between them. Therefore, we developed a new physics engine, named *FluidEngine*, to support simulating the tasks proposed in FluidLab. FluidEngine is a general-purpose physics simulator that supports modeling solid, liquid and gas, covering materials including elastic, plastic and rigid objects, Newtonian and non-Newtonian liquids, as well as smoke and air. FluidEngine is developed using Taichi (Hu et al., 2019b), a domain-specific language embedded in Python, and supports massive parallelism on GPUs. In addition, it is implemented in a fully *differentiable* manner, providing useful gradient information for downstream optimization tasks. FluidEngine follows the OpenAI Gym API (Brockman et al., 2016) and also comes with a user-friendly Python interface for building custom environments. We also provide a GPU-accelerated renderer developed with OpenGL, supporting realistic image rendering in real-time.

We evaluate state-of-the-art Reinforcement Learning algorithms and trajectory optimization methods in FluidLab, and highlight major challenges with existing methods for solving the proposed tasks. In addition, since differentiable physics has proved useful in many rigid-body and soft-body robotic tasks (Xu et al., 2021; Lin et al., 2022a; Xu et al., 2022; Li et al., 2022), it is desirable to extend it to fluid manipulation tasks. However, supporting such a wide spectrum of materials present in FluidLab in a differentiable way is extremely challenging, and optimization using the gradients is also difficult due to the highly non-smooth optimization landscapes of the proposed tasks. We propose several optimization schemes that are generally applicable to the context of fluids, making our simulation fully differentiable and providing useful directions to utilize gradients

for downstream optimizations tasks. We demonstrate that our proposed techniques, coupled with differentiable physics, could be effective in solving a set of challenging tasks, even in those with highly turbulent fluid (e.g., inviscid air flow). In addition, we demonstrate that the trajectories optimized in simulation are able to perform reasonably well in several example tasks, when directly deployed in real-world settings. We summarize our main contributions as follows:

- We propose *FluidLab*, a set of standardized manipulation tasks to help comprehensively study *complex fluid manipulation* with challenging task settings and potential real-world applications.

- We present *FluidEngine*, the underlying physics engine of FluidLab that models various material types required in FluidLab's tasks. To the best of our knowledge, it is the first-of-its-kind physics engine that supports such a wide spectrum of materials and couplings between them, while also being *fully differentiable*.

- We benchmark existing RL and optimization algorithms, highlighting challenges for existing methods, and propose optimization schemes that make differentiable physics useful in solving a set of challenging tasks in both simulation and real world. We hope FluidLab could benefit future research in developing better methods for solving complex fluid manipulation tasks.

## 2 RELATED WORK

**Robotic manipulation of rigid bodies, soft bodies and fluids** Robotic manipulation has been extensively studied over the past decades, while most of previous works concern tasks settings with mainly rigid objects (Xian et al., 2017; Suárez-Ruiz et al., 2018; Tung et al., 2020; Zeng et al., 2020; Chen et al., 2022). In recent years, there has been a rising interest in manipulation problems concerning systems with deformable objects, including elastic objects such as cloth (Liang et al., 2019; Wu et al., 2019; Lin et al., 2022b), as well as plastic objects like plasticine (Huang et al., 2021). Some of them also study manipulation problems involving fluids, but mainly consider relatively simple problems settings such as robotic pouring (Sieb et al., 2020; Narasimhan et al., 2022) or scooping (Antonova et al., 2022), and mostly deal with only water (Schenck & Fox, 2018; Ma et al., 2018; Seita et al., 2022; Antonova et al., 2022). In this work, we aim to study a comprehensive set of manipulation tasks involving different fluid types with various properties. It is also worth mentioning that aside from the robotics community, a vast literature has been devoted to fluid control in computer graphics (Hu et al., 2019a; Zhu et al., 2011; Yang et al., 2013), with their focus on artistic visual effects yet neglecting the physical feasibility (e.g., arbitrary control forces can be added to deform a fluid body into a target shape). In contrast to them, we study fluid manipulation from a realistic *robotics* perspective, where fluids are manipulated with an embodied robotic agent.

**Simulation environments for robotic policy learning** In recent years, simulation environments and standardized benchmarks have significantly facilitated research in robotics and policy learning. However, most of them (Brockman et al., 2016; Kolve et al., 2017; Xiang et al., 2020) use either PyBullet (Coumans & Bai, 2016) or MuJoCo (Todorov et al., 2012) as their underlying physics engines, which only supports rigid-body simulation. Recently, environments supporting soft-body manipulation, such as SAPIEN (Xiang et al., 2020), TDW (Gan et al., 2020) and SoftGym (Lin et al., 2020) adopted Nvidia FleX (Macklin et al., 2014) engine to simulate deformable objects and fluids. However, FleX lacks support for simulating plastic materials or air, doesn't provide gradient information, and its underlying Position-Based Dynamics (PBD) solver has been considered not physically accurate enough to produce faithful interactions between solids and fluids.

**Differentiable physics simulation** Machine Learning with differentiable physics has gained increasing attention recently. One line of research focuses on learning physics models with neural networks (Li et al., 2018; Pfaff et al., 2020; Xian et al., 2021), which are intrinsically differentiable and could be used for downstream planning and optimization. Learning these models assumes ground-truth physics data provided by simulators, hasn't proved to be capable of simulating a wide range of materials, and could suffer from out-of-domain generalization gap (Sanchez-Gonzalez et al., 2020). Another approach is to implement physics simulations in a differentiable manner, usually with automatic differentiation tools (Hu et al., 2019a; de Avila Belbute-Peres et al., 2018; Xu et al., 2022; Huang et al., 2021), and the gradient information provided by these simulations has shown to be helpful in control and manipulation tasks concerning rigid and soft bodies (Xu et al., 2021; Lin et al., 2022a; Xu et al., 2022; Li et al., 2022; Wang et al., 2023). On a related note, Taichi (Hu et al., 2019b) and Nvidia Warp (Macklin, 2022) are two recently proposed domain-specific programming language for GPU-accelerated simulation. FluidLab is implemented using Taichi.

## 3 FLUIDENGINE

### 3.1 SYSTEM OVERVIEW

FluidEngine is developed in Python and Taichi, and provides a set of user-friendly APIs for building simulation environments. At a higher level, it follows the standard OpenAI Gym API and is compatible with standard reinforcement learning and optimization algorithms. Each environment created using FluidEngine consists of five components: i) a robotic agent equipped with user-defined end-effector(s); ii) objects imported from external meshes and represented as signed distance fields (SDFs); iii) objects created either using shape primitives or external meshes, represented as particles; iv) gas fields (including a velocity field and other advected quantity fields such as smoke density and temperature) for simulating gaseous phenomena on Eulerian grids; and v) a set of user-defined geometric boundaries in support of sparse computations. After adding these components, a complete environment is built and enables inter-component communications for simulation. We encourage readers to refer to our project site for visualizations of a diverse set of scenes simulated with FluidEngine.

### 3.2 MATERIAL MODELS

FluidEngine supports various types of materials, mainly falling under three categories: solid, liquid and gas. For solid materials, we support rigid, elastic and plastic objects. For liquid, we support both Newtonian and non-Newtonian liquids. Both solid and liquid materials are represented as particles, simulated using the Moving Least Squares Material Point Method (MLS-MPM) (Hu et al., 2018), which is a hybrid Lagrangian-Eulerian method that uses particles and grids to model continuum materials. For gas such as smoke or air, we simulate them as incompressible fluid on a Cartersian grid using the advection-projection scheme (Stam, 1999).

**Elastic, plastic, and liquid** Both elastic and plastic materials are simulated following the original MLS-MPM formulation (Hu et al., 2018). For both materials, we compute the updated deformation gradient, internal force, and material stress in the particle-to-grid (P2G) step, and then enforce boundary conditions in grid operations. Corotated constitutive models are used for both materials. For plastic material, we additionally adopt the Box yield criterion (Stomakhin et al., 2013) to update deformations. We refer readers to the original MPM papers (Hu et al., 2018; Stomakhin et al., 2013) for more details. Both viscous and non-viscous liquids are also modeled using the corotated models, and we additionally reset the deformation gradients to diagonal matrices at each step. For non-Newtonian liquids such as ice cream, we model them using the elastoplastic continuum model with its plastic component treated by the von Mises yield criterion (following Huang et al. (2021)).

**Rigid body** Two types of geometric representations are used to support rigid bodies in FluidEngine. The first one is represented by particles. We simulate rigid objects by treating them as elastic objects first using MLS-MPM, then we compute an object-level transformation matrix and enforce the rigidity constraint. Another type of rigid object is represented using signed distance fields (SDFs), usually generated by importing external meshes. This representation is more computation-efficient since it computes its dynamics as a whole, generally used for physically big meshes.

**Gas** We develop a grid-based incompressible fluid simulator following (Stam, 1999) to model gaseous phenomena such as air and smoke. A particle-based method is not well suited in these cases due to its (weakly) compressible limit.

**Inter-material coupling** MPM naturally supports coupling between different materials and particles, computed in the grid operations. For interactions (collision contact) between mesh-based objects and particle-based materials, we represent meshes as time-varying SDFs and model the contact by computing surface normals of SDFs and applying Coulomb friction (Stomakhin et al., 2013). For interaction between mesh-based objects and the gas field, we treat grid cells occupied by meshes as the boundaries of the gas field, and impose boundary conditions in the grid operations. Coupling between gas and particle-based materials is also implemented at the grid level, by computing a impact on the gas field based on the particle velocity field.

### 3.3 DIFFERENTIABILITY AND RENDERING

FluidEngine is implemented based on Taichi's autodiff system to compute gradients for simple operations. We further implement our own gradient computations for complex operations such as SVD and the projection step in the gas simulation, and efficient gradient-checkpointing to allow gradient

flow over long temporal horizons, unconstrained by GPU memory size. FludiEngine also comes with a real-time OpenGL-based renderer. More details are in Appendix A.

## 3.4 COMPARISON WITH OTHER SIMULATION ENVIRONMENTS

| Simulators | Differentiable | Solid | | | Liquid | | Gas |
|---|---|---|---|---|---|---|---|
| | | Rigid | Elastic | Plastic | Newt. | Non-Newt. | Air / Smoke |
| PyBullet | | ✓ | ✓ | | | | |
| MuJoCo | | ✓ | | | | | |
| FleX | | ✓ | ✓ | | ✓ | | ✓[XB] |
| SoftGym | | ✓ | ✓ | | ✓ | | |
| PlasticineLab | ✓[XA] | ✓ | | ✓ | | | |
| SAPIEN | | ✓ | ✓ | | ✓ | | |
| TDW | | ✓ | ✓ | | ✓ | | |
| DEDO | | ✓ | ✓ | | | | |
| DiffSim | ✓ | ✓ | ✓ | ✓ | | | |
| DiSECt | ✓ | ✓ | ✓ | ✓ | | | |
| FluidLab | ✓ | ✓ | ✓ | ✓ | ✓ | ✓ | ✓ |

Table 1: **Comparison with other popular simulators.** [A]Plasticine lab offers differentiable simulation, but doesn't provide gradient checkpointing, hence only supporting gradient flow over relatively short horizons. [B]FleX demonstrates the capability of smoke simulation in their demo video, but this feature and smoke rendering are not available in their released codebase.

There exist many simulation engines and environments for robotics research, and here we compare FluidLab and FluidEngine with some popular ones among them, including PyBullet (Coumans & Bai, 2016), MuJoCo (Todorov et al., 2012), FleX (Macklin et al., 2014), SoftGym (Lin et al., 2020), PlasticineLab (Huang et al., 2021), SAPIEN (Xiang et al., 2020), TDW (Gan et al., 2020), DiffSim (Qiao et al., 2020), DiSECt (Heiden et al., 2021), and DEDO (Antonova et al., 2021). Table 1 shows a comparison in terms of differentiability and supported material types. FluidLab covers a wider range of materials compared to existing environments. In addition, many of these environments rely on physics engines written in C++, and some are not open-sourced (such as FleX), making interpretability and extensibility a major hurdle, while FluidLab is developed in Python and Taichi, enjoying both GPU-acceleration and user-extensibility. In order to validate the accuracy and reliability of FluidEngine, we conducted a number of validation simulations, as shown in Appendix E. We also present a list of detailed comparisons with relevant differentiable simulators in Appendix F.

## 4 FLUIDLAB MANIPULATION TASKS

FluidLab presents a collection of complex fluid manipulation tasks with different fluid materials and their interactions with other objects. In general, we observe that manipulations tasks involving fluids can be categorized into two main groups: i) where fluid functions as a *transmitting media* and is used to manipulate other objects which are otherwise unreachable, and ii) where fluid is the *manipulation target* and the goal is to move or deform the fluid volume to a desired configuration. We present a suite of 10 different tasks in FluidLab, as shown in Figure 2, where in each task an agent needs to manipulate or interact with the fluids in the scene to reach a specific goal configuration, falling under the aforementioned two categories.

### 4.1 TASK DETAILS

FluidLab contains the following 10 different manipulation tasks presented in Figure 2, all of which are motivated by real-world scenarios where an agent interacts with certain types of fluids. We additionally discuss the reasoning behind the choices of our task selections in Appendix G.

**Fined Grained Pouring** A glass filled with two types of non-Newtonian liquid with different densities, colored as blue and white respectively. The task is to pour the light blue liquid out of the glass while retaining as much white heavy liquid as possible.

**Gathering (Easy & Hard)** An agent equipped with a rigid spatula is able to paddle the water in a tank, but its motion is restricted within the green area. The goal is to make water currents so that the two floating objects will be gathered towards the marked goal location. The easy setting has a

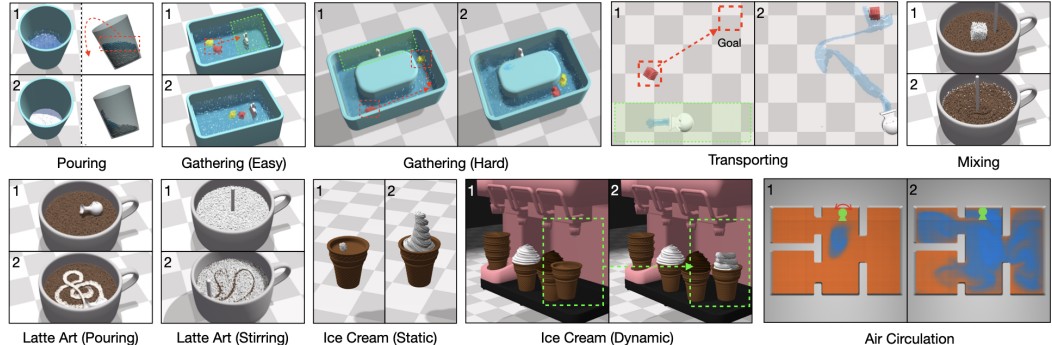

Figure 2: 10 Fluid manipulation tasks proposed in FluidLab. In each task, 1 depicts the initial condition and 2 shows a desired goal configuration. Both top and side views are shown for Pouring.

relatively open space, while in the hard setting the spatial configuration is more complex and the two objects need to move in different directions.

**Transporting** A water jet bot is required to move an object of interest to a pre-determined goal location. The agent is only allowed to move within the green area, and thus can only transport the object using the water it injects.

**Mixing** Given a cube of sugar put in a cup of coffee, the agent needs to plan generate sequence of stirring action to mix the sugar particles with the coffee in the shortest period of time.

**Latte Art (Pouring & Stirring)** We consider two types of latte art making tasks: the first one is making latte art via pouring, where an agent pours milk into a cup of coffee to form a certain pattern; the second one is via stirring, where a layer of viscous frothed milk covers a cup of coffee, and the agent needs to manipulate a rigid stick to stir in the cup to reach a specified goal pattern.

**Ice Cream (Static & Dynamic)** This task involves policy learning concerning a non-Newtonian fluid – ice cream. We consider two settings here. The first one is a relatively simple and static setting, where we control an ice cream emitter that moves freely in space and emits ice cream into a cone and form a desired spiral shape. The second setting is more dynamic, where ice cream is squeezed out from a machine and an agent controls the biscuit cone. During the process, the agent needs to reason about the dynamics of the icecream and its interaction with the cone since the ice cream moves together with the cone due to friction.

**Air Circulation** The agent controls the orientation of an air conditioner (denoted in green in the figure) which produces cool air (colored in blue) in a house with multiple rooms, initially filled with warm air (colored in orange), and the goal is to cool down only the upper left and the right room, while maintaining the original temperature in the lower left room. Completing this challenging task requires fine-grained control and reasoning over the motion of the cooling air.

## 4.2 TASK AND ACTION REPRESENTATIONS

We model each task as a finite-horizon Markov Decision Process (MDP). An MDP consists of a state space $\mathcal{S}$, an action space $\mathcal{A}$, a transition function $\mathcal{T} : \mathcal{S} \times \mathcal{A} \to \mathcal{S}$, and a reward function associated with each transition step $\mathcal{R} : \mathcal{S} \times \mathcal{A} \times \mathcal{S} \to \mathbb{R}$. The transition function is determined by the physics simulation in FluidLab. We discuss the loss and reward for each task used for experiments in Section 5. The goal of the agent in the environment is to find a policy $\pi(a|s)$ that produces action sequences to maximize the expected total return $E_\pi[\Sigma_{t=0}^T \gamma^t \mathcal{R}(s_t, a_t)]$, where $\gamma \in (0, 1)$ denotes the discount factor and $T$ is the horizon of the environment.

**State, Observation, and Action Space** We represent MPM-based materials with a particle-based representation, using a $N_p \times d_p$ state matrix, where $N_p$ is the number of all particles in the system, and $d_p = 6$ contains the position and velocity information of each particle. To represent the gas field, which is simulated using a grid-based representation, we use another $N_c \times d_c$ matrix to store information of all grid cells, where $N_c$ is the number of cells in the grid, and $d_c$ is a task-dependent vector. In the air circulation task, $d_c = 4$ contains the velocity (a 3-vector) and temperature (a scalar) of each cell. In addition, the state also includes the 6D pose and velocity (translational and rotational) information of the end-effector of the robotic agent in the scene. All the tasks use an MPM grid of size $64^3$, with around 100,000 particles, and a $128^3$ grid for gas simulation. The full action space of FluidLab is a 6-vector, containing both translational and angular velocity of the

agent's end-effector. For more details of our task settings, their simulations, and the input to RL policies, please refer to Appendix B.1.

## 5 EXPERIMENTS

We evaluate model-free RL algorithms, sampling-based optimization methods as well as trajectory optimization using differentiable physics coupled with our proposed optimization techniques in FluidLab. We discuss our optimization techniques and result findings in this Section. For detailed loss and reward design, please refer to Appendix B.2.

### 5.1 TECHNIQUES AND OPTIMIZATION SCHEMES USING DIFFERENTIABLE PHYSICS

Differentiable simulation provides useful gradient information that can be leveraged for policy optimization. However, oftentimes the dynamics function could both be highly non-smooth (e.g. in case of a contact collision, where the gradient is hard to obtain), and exhibit a complex optimization landscape where the optimizer can easily get trapped in local minima. We discuss a few techniques and optimization schemes used in our experiments here for obtaining more informative gradients and leveraging them more effectively in the context of fluid manipulation.

**Soft Contact Model** In our simulation, the contact between particle-based fluid bodies and SDF-based rigid objects occurs in the grid operations. Such contact induces drastic state changes during the classical MPM process. We adopt a softened contact model in our simulation: we consider contact occurs when the particles and the rigid objects are within a certain distance threshold $\epsilon$, and compute an after-collision velocity $v_c$. Afterwards, we blend this velocity with the original velocity of the objects using a distance-based blending factor $\alpha$, given by $v_{\text{new}} = \alpha v_c + (1 - \alpha)v_{\text{original}}$, where $\alpha = \min\{\exp(-d), 1\}$ and $d$ is the distance between the two objects. This provides a smoothed contact zone and thus offers smoothed gradient information during contact.

**Temporally Expanding Optimization Region** During gradient-based trajectory optimization, if we consider the full loss computed using the whole trajectory with respect to a certain goal configuration, the whole optimization landscape could be highly nonconvex and results in unstable optimization. In our experiments, we warm start our optimization with a restricted temporal region of the trajectory and keeps expanding this region steadily whenever a stabilized optimization spot is reached for the current region. This greatly helps stabilize the optimization process using differentiable physics and produces better empirical results, as we show in Section 5.2.

**Gradient Sharing within Fluid Bodies** Many optimization scenarios involving fluids could be highly non-convex. Consider a simple pouring task where the goal is to pour all the water from a mug, with the loss defined as the distance between the ground floor and all the water particles. During the exploration phase, if a water particle was never poured out of the mug, it is trapped in a local minimum and the loss gradient won't be able to guide towards a successful pouring. Thanks to the continuity of fluid materials, we are able to share gradients between adjacent particles in a fluid body and use those informative gradients to guide other particles. Such gradient sharing can be achieved by updating the gradient on each particle via accumulating neighboring particle gradients as follows: $\frac{\partial \mathcal{L}}{\partial q_i} = \Sigma_j w(\mathcal{L}_j, d_{ij})\frac{\partial \mathcal{L}}{\partial q_j}$, where $j$ iterates over all neighboring particles of $i$, $q$ represents particle properties such as velocity or position, and $w$ is a kernel function conditioned on both the distance between $i$ and $j$ and the loss of particle $j$, assigning a higher weight to neighboring particles who incurred a lower loss at the previous iteration. Note that in practice, such gradient sharing requires nearest neighbour search for all particles at each timestep, which could be computationally slow. Therefore, we employ an additional attraction loss to approximate gradient sharing, where particles that incurred a lower loss at the previous iteration will attract other particles around it: $\mathcal{L}_{att} = \Sigma_i \Sigma_j w(\mathcal{L}_j, d_{ij})\|p_i - p_j\|$.

### 5.2 METHOD EVALUATION WITH FLUIDLAB TASKS

We evaluate our proposed optimization schemes coupled with differentiable physics (DP), model-free RL algorithms including Soft Actor-Critic (SAC) (Haarnoja et al., 2018) and Proximal Policy Optimization (PPO) (Schulman et al., 2017), CMA-ES (Hansen & Ostermeier, 2001), a sampling-based trajectory optimization method, as well as PODS (Mora et al., 2021) an method combines RL and differentiable simulation (see Appendix C for a discussion). We use the open source implementation (Raffin et al., 2021) for PPO and SAC, and pycma (Hansen et al., 2019) for CMA-ES.

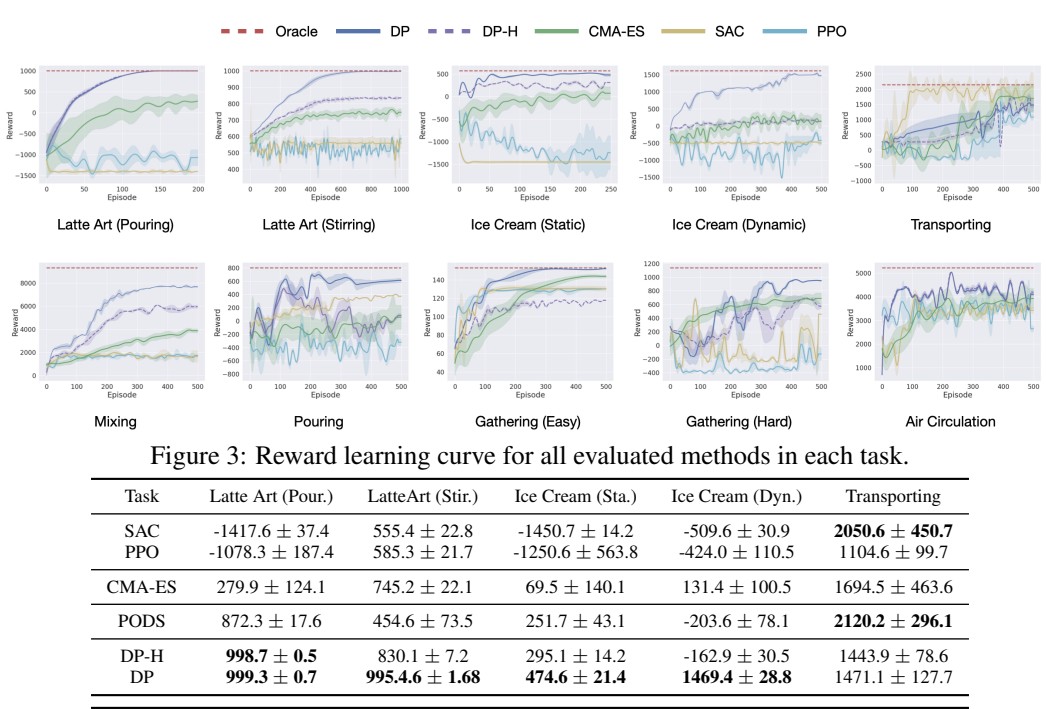

Figure 3: Reward learning curve for all evaluated methods in each task.

| Task | Latte Art (Pour.) | LatteArt (Stir.) | Ice Cream (Sta.) | Ice Cream (Dyn.) | Transporting |
|---|---|---|---|---|---|
| SAC | -1417.6 ± 37.4 | 555.4 ± 22.8 | -1450.7 ± 14.2 | -509.6 ± 30.9 | **2050.6 ± 450.7** |
| PPO | -1078.3 ± 187.4 | 585.3 ± 21.7 | -1250.6 ± 563.8 | -424.0 ± 110.5 | 1104.6 ± 99.7 |
| CMA-ES | 279.9 ± 124.1 | 745.2 ± 22.1 | 69.5 ± 140.1 | 131.4 ± 100.5 | 1694.5 ± 463.6 |
| PODS | 872.3 ± 17.6 | 454.6 ± 73.5 | 251.7 ± 43.1 | -203.6 ± 78.1 | **2120.2 ± 296.1** |
| DP-H | **998.7 ± 0.5** | 830.1 ± 7.2 | 295.1 ± 14.2 | -162.9 ± 30.5 | 1443.9 ± 78.6 |
| DP | **999.3 ± 0.7** | **995.4.6 ± 1.68** | **474.6 ± 21.4** | **1469.4 ± 28.8** | 1471.1 ± 127.7 |

| Task | Mixing | Pouring | Gathering (Easy) | Gathering (Hard) | Air Circulation |
|---|---|---|---|---|---|
| SAC | 1523.2 ± 389.4 | 374.5 ± 7.3 | 130.2 ± 2.4 | 477.2 ± 9.7 | 4430.2 ± 264.7 |
| PPO | 1728.3 ± 159.4 | -324.3 ± 50.3 | 129.7 ± 1.2 | -138.9 ± 67.3 | 4282.2 ± 610.9 |
| CMA-ES | 3875.2 ± 143.2 | 57.1 ± 169.7 | 143.1 ± 1.4 | 687.2 ± 80.1 | 3917.3 ± 385.3 |
| PODS | 4742.3 ± 187.6 | 284.1 ± 84.9 | 129.5 ± 1.7 | 413.4 ± 71.6 | 3557.8 ± 364.2 |
| DP-H | 5974.0 ± 198.8 | 88.3 ± 9.1 | 116.9 ± 0.3 | 566.9 ± 30.4 | **5046.3 ± 23.7** |
| DP | **7648.4 ± 47.0** | **603.4 ± 29.7** | **151.5 ± 0.3** | **946.8 ± 5.1** | 5043.3 ± 21.2 |

Table 2: The final accumulated reward and the standard deviation for each evaluated method.

Additionally, we choose to optimize for periodic policies for the two Gathering tasks and the Mixing task, where the whole task horizon is evenly divided into multiple periods, and each period contains an optimizable action trajectory followed by a reseting action moving the end-effector back to a neutral position. We use the same accumulated reward to evaluate all the methods. Additionally, we include the performance of an oracle policy controlled by a human operator as a reference. The reward learning curves are shown in Figure 3, and we report the final performance of all methods upon convergence in Table 2. Please refer to our project page for more qualitative results.

**Model-free RL Algorithms** RL algorithms demonstrate reasonable learning behaviors in tasks with relatively small observation spaces, such as Transporting (where only a subset of water particles are present in the scene at each step) and Air Circulation. However, in environments such as Latte Art, Mixing, and Ice Cream, RL methods struggle to generate a reasonable policy. We suspect that this is because 1) the high-dimensional space used to describe the fluid system, which is hard for RL algorithms to abstract away useful information about the system within limited exploration iterations and training budget, making it less sample efficient than DP, and 2) the non-linear dynamics of the complex fluids such as viscous frothed milk and the non-Newtonian ice creams, as well as their interactions with other solid materials, make policy search of RL policies based on sampled actions extremely difficult since randomly sampled actions and rewards are not informative enough in conveying contact and deformation information about the fluid systems.

**Sampling-based trajectory optimization** CMA-ES does reasonably well in many tasks with simple spatial configuration as goals. However, it struggles in fine-grained shape matching tasks (*e.g.* Ice Cream) and fine-grained control tasks (*e.g.* pouring). This is because CMA-ES searches for action sequences via trajectory sampling, and thus cannot handle delicate interactions well.

**Differentiable physics-based trajectory optimization** In most of the tasks, the gradients provided by our differentiable simulation are able to guide trajectory optimization towards a good optimization spot, resulting in near-oracle performance. It also demonstrates better sample-efficiency than

other methods in most tasks. We additionally compare with a *hard* version of differentiable physics (DP-H), which uses the gradient to optimize trajectory, but without using the optimization techniques proposed in Section 5.1. DP-H does similarly well on tasks with relatively simple settings, but fails to produce good trajectories in scenarios with interactions between fluids and other materials bodies, as well as in tasks with more complex and non-convex optimization landscapes, such as Pouring with fine-grained control objective. The experimental results demonstrate that the gradient information provided by our differentiable simulator, coupled with the optimization schemes we proposed, provides informative optimization information when dealing with complex fluid systems.

**Sim-to-real transfer** We additionally evaluate the optimized trajectories in a real-world robotic setup and show that the learned trajectories can perform the desired tasks reasonably well qualitatively in the real world. See Appendix D for more details.

## 5.3 Discussions & Potential Future Research Directions

In our experiments above, we evaluated a range of policy learning and trajectory optimization methods. Our results show that when coupled with our proposed optimization scheme, differentiable physics can provide useful gradient information to guide trajectory optimization, and show reasonable performance in a range of complex fluid manipulation task settings, even in those with fine-grained shape matching goals and highly unstable fluid systems such as air flow. Meanwhile, our results suggested a number of major challenges in solving complex fluid manipulation tasks. First of all, the model free RL methods and also the method combining RL and differentiable simulation for policy learning struggle to produce competitive results. We believe one main reason for this is it's difficult for the policy to informatively interpret the high-dimensional state of the complex fluid systems. It's worth investigating what would be an efficient representation for learning such closed-loop policies: point cloud-based representation or image input could be a good starting point, and more abstract representations with semantic understanding of the material properties and appearances are also worth studying. Second, one clear goal is to combine our optimization scheme with policy learning, and to distill optimized trajectories using differentiable physics into more general close-loop control policies. Thirdly, although our current task selections cover a range of different scenarios and materials, they are still relatively short-horizon, in the sense that there's usually only a single action mode. It would be very interesting to extend task design to more realistic latte-art making, which requires action switching between both pouring and stirring. One promising direction is to learn a more abstract dynamics model of the fluid system and use it for long-horizon action planning and skill selection. Also, our current latte art tasks rely on dense reward signal computed from a temporal trajectory of goal patterns, and a more abstract and learned dynamics model would probably allow optimization using a static goal pattern. Last but not least, our real world experiments show that there's still a perceivable gap between our simulated material behavior and the real world environments. Improving both simulation performance and close-loop policy learning are needed to enable robots to perform better in more complex tasks, such as more realistic latte art making.

## 6 Conclusion and Future Work

We presented FluidLab, a virtual environment that covers a versatile collection of complex fluid manipulation tasks inspired by real-world scenarios. We also introduced FluidEngine, a backend physics engine that can support simulating a wide range of materials with distinct properties, and also being fully differentiable. The proposed tasks enabled us to study manipulation problems with fluids comprehensively, and shed light on challenges of existing methods, and the potential of utilizing differentiable physics as an effective tool for trajectory optimization in the context of complex fluid manipulation. One future direction is to extend towards more realistic problem setups using visual input, and to distill policy optimized using differentiable physics into neural-network based policies (Lin et al., 2022a; Xu et al., 2023), or use gradients provided by differentiable simulation to guide policy learning (Xu et al., 2022). Furthermore, since MPM was originally proposed, researchers have been using it to simulate various realistic materials with intricate properties, such as snow, sand, mud, and granular materials. These more sophisticated materials can be seamlessly integrated into the MPM simulation pipeline in FluidLab, and will be helpful for studying manipulation tasks dealing with more diverse set of materials. We believe our standardized task suite and the general purpose simulation capability of FluidLab will benefit future research in robotic manipulation involving fluids and other deformable materials.

ACKNOWLEDGMENTS

We would like to thank Qi Wu for helping with conducting the real-world ice cream experiments, Pengsheng Guo for helping with implementing the volumetric smoke renderer, and Jiangshan Tian for teaching the authors everything about making latte art. We would also like to thank Zhiao Huang, Xuan Li, and Tsun-Hsuan Wang for their constructive suggestions during the project development. This material is based upon work supported by MIT-IBM Watson AI Lab, Sony AI, a DARPA Young Investigator Award, an NSF CAREER award, an AFOSR Young Investigator Award, DARPA Machine Common Sense, National Science Foundation under Grant No. 1919647, 2106733, 2144806, and 2153560, and gift funding from MERL, Cisco, and Amazon.

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

## A    Details on FluidLab's Differentibility and Rendering

**Differentiability and gradient checkpointing** We use the automatic differentiation tool (autodiff) provided by Taichi to get gradients for most of our computations; for more complex computations and kernels not supported by Taichi's autodiff, we implemeneted gradient computation ourselves, including SVD computations and the projection step in gas simulation. One additional challenge for making the whole simulation fully differentiable is enabling gradient flow through time, which requires bookkeeping system states of all timesteps. Since most of FluidLab's tasks have tens of thousands of simulation steps, it is impossible to fit the whole computation graph into the GPU memory. In order to address this, we implemented efficient gradient checkpointing to enable gradient flow over unlimited time horizons. During the forward process of the simulation, only a local trajectory of the environment states are stored in the GPU memory for fast computation. Whenever the GPU memory is full, we store the latest state of the environment to a caching device (either CPU memory or local disks), and clear up the GPU memory for further simulation. Once a complete forward pass is done, we restore the cached states and run forward simulation again to fulfill the GPU memory with the local chunk of simulation trajectory. Note that instead of caching all intermediate states, we only store one single step of the states at every caching step, and run simulation again to restore the states of the local trajectory; therefore, the i/o step of such caching operations is very efficient. This enables us to propagate loss and gradient information backward in time through a simulation trajectory with arbitrary number of steps, unconstrained by the GPU memory size.

**Rendering** We implemented a photo-realistic renderer for visualizing scenes created with FluidEngine. The renderer is developed in C++ using OpenGL, supports GPU-accelerated real-time rendering. Part of our rendering pipeline was modified based on FleX's rendering system, enhanced with various features such as particle-level colorization, headless rendering, dynamic object loading, volume rendering for smoke field, etc. We also provided a set of python APIs to dynamically update the scene configuration from within Python. The renderer supports rendering materials as either particles or fluids. Particle-based rendering is more interpretable for visualization and debugging purposes, while fluid-based rendering is more photo-realistic and allows potential image-based sim-to-real transfer applications in the future.

## B    FluidLab Task and Evaluation Details

### B.1    Task Details

For RL algorithms, we downsample the number of particles and cells for each fluid body in the scene to avoid a intractably huge observation space. The state matrix is then flattened and fed into an MLP policy network. We do not evaluate image-based RL methods as many tasks FluidLab are 3D and occlusion could be a potential problem, and leave such evaluations for future work.

All our tasks use a simulation step of 2e-3 seconds, each containing 10 substeps of 1e-4 seconds for ensuring simulation stability. All our proposed tasks contain around 100,000 particles and runs in real time (60 FPS where each frame is a simulation step) on a desktop computer equipped with an Nvidia RTX 3090 GPU and an Intel i7-8700K CPU, and the typical GPU usage is under 50%.

Each task has a task-specific action range and action space, with certain dimensions of the full action-space locked. We present our task-specific settings for action spaces, observed number of particles per fluid body and observed number of grid cells in Table 3. Detailed properties of all materials used are listed in Table 4.

### B.2    Loss and Reward

**Latte Art (Pouring)**, **Latte Art (Stirring)**, **Ice Cream (Static)** and **Ice Cream (Dynamic)**: These tasks have a set of predefined goal patterns, and the loss is defined by summing up the Chamfer distance between particles in the current state of the scene and the particles in the goal pattern at each time step. Ideally, the goal pattern should be a static goal shape of the final step, but using a single goal pattern would be extremely challenging for RL methods: if only the last step of the trajectory is used for reward computation, such an extremely sparse reward will make learning infeasible; computing a dense reward based on the distance between the current state of each timestep and the

| Task | Action space | | number of particles | number of grid cells |
| --- | --- | --- | --- | --- |
| | Trans. Velocity | Rot. Velocity | per fluid body | for gas field |
| Latte Art (Pouring) | Along x and z | - | 1000 | - |
| Latte Art (Stirring) | Along x and z | - | 1000 | - |
| Ice Cream (Static) | Along x, y and z | - | 2000 | - |
| Ice Cream (Dynamic) | Along x, y and z | - | 2000 | - |
| Transporting | Along x and z | Around y | 500 | - |
| Mixing | Along x and z | - | 1000 | - |
| Pouring | - | Around z | 500 | - |
| Gathering (Easy) | Along x and z | - | 500 | - |
| Gathering (Hard) | Along x and z | - | 500 | - |
| Air Circulation | - | Around y | - | $128 \times 9 \times 128$ |

Table 3: Task-specfic settings.

| Material | Type | Lamé parameters | | Density $\rho$ |
| --- | --- | --- | --- | --- |
| | | $\mu$ | $\lambda$ | |
| Water | Liquid | 0.0 | 277.78 | 1.0 |
| Milk | Liquid | 0.0 | 277.78 | 1.0 |
| Frothed Milk | Viscous Liquid | 208.33 | 277.78 | 1.0 |
| Coffee | Liquid | 0.0 | 277.78 | 1.0 |
| Ice Cream | Non-newtonian Liquid | 416.67 | 277.78 | 0.5 |
| Floating object | Elastic | 416.67 | 277.78 | 0.5 |
| Sugar cube | Viscous Liquid | 208.33 | 277.78 | 1.0 |
| Light Liquid in Pouring | Liquid | 0.0 | 277.78 | 0.8 |
| Heavy Liquid in Pouring | Liquid | 0.0 | 277.78 | 1.5 |
| Object in Transporting | Rigid | 416.67 | 277.78 | 5.0 |

Table 4: Details of materials properties used in FluidLab.

goal pattern will also be non-informative: due to the dynamic movement within the fluid bodies, mimicing the static goal pattern at each timestep will not result in successful learning. Therefore, in order to make an informative comparison, we use a temporal trajectory of the desired goal pattern to compute the loss $L = \Sigma_t \mathrm{CD}(s_t, s_t^{goal})$, where CD denotes Chamfer Distance.

**Transporting**, **Pouring**, **Gathering (Easy)** and **Gathering (Hard)**: These tasks use a single spatial location as the goal and the loss is computed as: $\mathcal{L} = \Sigma_i \|p_i - p_{goal}\|$, which is summed over all particles of the object(s) to be manipulated. For **Pouring**, we apply this loss between the particles of the upper layer fluid and the ground location, with an additional loss attracting the particles of the lower layer fluid to their initial positions.

**Mixing** The goal of this task is to mix the sugar particles with the coffee. The loss is designed to maximize the spatial coverage of the sugar particles and is defined as $\mathcal{L} = -\Sigma_i \Sigma_j \|p_i - p_j\|$, where both $i$ and $j$ iterate over all particles originally belonging to the sugar cube.

**Air circulation** We place 27 sensors in the house, with 9 sensors in each room uniformly. The loss is defined as $\mathcal{L} = \Sigma_{\mathrm{ul}} |T_{\mathrm{ul}} - T_{\mathrm{cool}}| + \Sigma_{\mathrm{r}} |T_{\mathrm{r}} - T_{cool}| + \Sigma_{\mathrm{ll}} |T_{\mathrm{ll}} - T_{\mathrm{warm}}|$, where $T$ represents temperature, and `ul`, `r` and `ll` denote sensors in the upper left, right and lower left room respectively.

We use the losses discussed above for optimization-based methods; for model-free RL methods, the reward of each task is simply defined as $\mathcal{R} = c_1 - c_2\mathcal{L}$, where $c_1$ and $c_2$ are constants to ensure a reasonable reward scaling for each task.

## C  EVALUATION OF PODS

We additionally evaluate PODS, a method that combines differentiable simulation and RL. It performs well in relatively convex tasks such as Latte Art (Pouring), and even outperforms DP and DP-H on simple tasks like transporting, mainly due to its exploratory nature from RL. However, it is not able to match performance in other complex tasks, especially in those where DP shows an clear advantage over DP-H and RL methods. We believe this is because PODS only uses local gradient information without our proposed optimization techniques; also, it is additionally learning a closed-loop policy which maps observation to action, where DP is only optimizing a trajectory. In fact, what PODS is learning aligns with the next step of our research, where a better scene representation to describe the fluid system is needed to better distill DP-optimized trajectory to a closed-loop policy network. Comparison with PODS suggests that a future research direction is how to encode the scene in a more informative representation for better policy learning, such as a pointcloud based representation; in addition, we expect better performance by combining our proposed optimization landscapes with methods that use both differentiable simulation and RL. These remain as our future work.

## D  SIM-TO-REAL TRANSFER

Our simulation platform provides an efficient test bed and useful gradient information for developing algorithms to deal with complex fluid manipulation problems, testing how the optimized policies perform in a real-world setting would be very helpful and informative in evaluating the sim-to-real gap and suggesting future research directions. Therefore, we conducted experiments on a real robot system, with a 7-DoF Franka Emika robot equipped with a parallel jaw gripper. We picked four representative tasks proposed in FluidLab, including Latte Art (Stirring), Ice Cream (Dynamic), Gathering and Mixing, and set up corresponding real world scenarios. We optimize the trajectories in simulation, and then command the robot to execute them using velocity control in the corresponding real-world scenarios. (See our project site for qualitative results.) It can be observed that although there's certain sim-to-real gap in terms of material behavior and dynamics, when we apply the simulation-optimized trajectories to real-world, the tasks can be completed to a reasonable extent. The sim-to-real gap mainly comes from the simulation inaccuracy and difference in material properties between sim and real. For example, in the latte art experiment, the frothed milk behaves a bit differently than in simulation: it's more sticky and tend to mix again after the latte art pen passes. This could be further improved in simulation by developing more accurate material models, as well as increasing the simulation resolution and number of particles, which we leave to our future work. Another example is the ice cream experiment, where it is difficult to maintain a steady flow speed of the ice cream, resulting in slight different end shape of the ice cream compared to the simulated one. Although there exists such sim-to-real gap in material dynamics, the robot is able to complete the proposed tasks reasonably well. We would also like to acknowledge that given our current resources, we are not able to conduct all the proposed tasks in real world, such as the air circulation task, which remains as our future work. However, We believe the current 4 selected tasks cover a representative range of materials and task settings, and such experiments could shed light on the value of our simulator as a test bed for robotic research, point out potential sim-to-real gap and suggest possible improvements in our future work.

## E  VALIDATION EXPERIMENTS FOR FLUIDENGINE

One useful way of validating the correctness of simulation engines is to run simulation experiments and verify if the produced simulation results match known physical phenomena. Below we list a number of classical physical phenomena that are typically observable in rigid body and computational fluid dynamics. We testified our simulation engine by running simulations to see if it can successfully reproduce the listed behaviors. Visual results of such simulations are included in our project website.

- Kármán vortex street: When a flow of fluid is separated by blunt bodies, a repeating pattern of swirling vortices would appear due to vortex shedding. We simulate a flow of fluid flowing through a rigid cylinder and visualize the top-down view. We colorized the fluid around the cylinder, and it can be seen that our simulator can faithfully produce a street of vortices.

- Magnus effect: This is an observable phenomenon commonly associated with a spinning object moving through a fluid. The path of the spinning object is deflected in a manner not present when the object is not spinning. One example of such phenomenon seen in daily life is the screw shot in football playing. This is an observable phenomenon commonly associated with a spinning object moving through a fluid. The path of the spinning object is deflected in a manner not present when the object is not spinning. One example of such phenomenon seen in daily life is the screw shot in football playing. Here we simulate a spinning ball in a fluid body. It can be observed that the spinning motion produces a deflection in the motion trajectory of the ball, and the extent of the deflection is dependent on its spinning velocity.

- Buoyancy: We verify here if objects with different density in a liquid body would incur different magnitude of buoyancy. It can be observed that the red all would float on the water surface due to buoyancy.

- Incompressibility and stable volume: Liquids are generally considered incompressible. Our MPM-based engine simulates weakly-compressible liquids, where the momentum and mass are updated every step via consecutive grid-particle-grid information passes. Therefore, one concern is if the simulated fluid bodies can maintain consistent pressure and volume through long-horizon simulations, without momentum and pressure loss. We verified our simulator can simulate a stable fluid body that maintains steady volume and pressure after 10,000 simulation steps.

- Conservation of momentum: When objects are in collision, their motions obey conservation of momentum. Here we test the collision of two objects with exactly the same mass on a friction-less floor, where it can be observed that they exchanged their momentum after the collision as expected.

- Varying bouncing behaviors with varying plasticity: Here we show different behaviors of the object bouncing on the floor when we vary the plasticity of the object. From left to right, we increase the plasticity of the object. Our simulator can simulate bouncing behavior with varying plasticity, where it can be observed energy dissipates faster with a greater plasticity.

- Rayleigh–Taylor instability: This phenomenon describes the instability of an interface between two fluids with different densities, which occurs when the heavy component is above and pushing the light one. Here we simulate the behavior of two layers of fluids with different densities, where it can be observed that a plume of heavy fluid with small vortices emerges and evolves due to the interfacial instabilities.

- Dam break: This is a classical engineering test case, where a fluid volume falls due to gravity and splash within the domain. We successfully reproduce this phenomenon with our simulation engine.

Our simulators successfully reproduced these physical phenomena, as we show on our project site. We believe running such validation experiments could informatively help demonstrate the accuracy and reliability of our simulation engine.

## F    COMPARISON WITH OTHER DIFFERENTIABLE SIMULATORS

### F.1    SPNETS

SPNets (Schenck & Fox, 2018) is a pioneering work in robotic manipulation of fluids. It is essentially a fluid simulator implemented as neural network layers, which makes it differentiable. Therefore, it is more like a differentiable simulator itself rather than a policy learning method. The parameters of SPNets are the physical parameters of the fluids. One limitation of SPNets is that its simulation capability: it only supports simulating single-phase non-Newtonian fluids, and only supports one-way coupling from static boundaries to the fluids, i.e. it doesn't support computing effects of fluids on other objects. In this work, one of our key insights is that in order to study realistic fluid manipulation problems that are motivated by real-world scenarios with realistic complexity, it's crucial to be able to support a wide range of materials and their inter-coupling. Our motivation is to

| | Material type | Contact with manipulator | Role of fluids | Key challenge / unique characteristic |
|---|---|---|---|---|
| Latte Art (Pouring) | Inviscid and viscous liquid | Indirect | Manipulation target | Coupling between different liquids |
| Latte Art (Stirring) | Inviscid and viscous liquid | Direct | Manipulation target | Coupling between liquids and the rigid manipulator |
| Ice Cream (Static) | Non-Newtonion fluid | Indirect | Manipulation target | Fine grained shape matching of non-Newtonian fluid |
| Ice Cream (Dynamic) | Non-Newtonion fluid | Direct | Manipulation target | Additional interaction with the rigid manipulator |
| Transporting | Inviscid liquid and rigid solid | Indirect | Transmitting medium | Use liquid to indirectly manipulate distant objects |
| Mixing | Inviscid and viscous liquid | Direct | Manipulation target | Coupling between different liquid body |
| Pouring | Inviscid liquid | Direct | Manipulation target | Fine-grained goal configuration |
| Gathering (Easy) | liquid and elastic solid | Direct | Transmitting medium | Use liquid to indirectly manipulate distant objects |
| Gathering (Hard) | liquid and elastic solid | Direct | Transmitting medium | More complex environment configuration |
| Air Circulation | Gas | Indirect | Manipulation target | Involving dynamics of highly unstable air flow |

Table 5: Comparison and characteristics of our selected tasks.

study problems beyond the single-phase water tasks proposed in SPNets, such as single-phase water pouring and catching, where only the effect of manipulators on the water is considered. Therefore, on the simulator side, SPNets is not able to simulate most of our propose tasks. In addition, on the task solving side, in the SPNets paper the author did propose to use gradients provided by the differentiable simulation to optimize trajectories for liquid control. Which is similar to our differentible physics based optimization method (DP-H), except that we proposed a few optimization techniques (DP) to help solving the tasks.

DiffSim (Qiao et al., 2020) is a differentiable simulation platform that leverages mesh-based representation and implicit differentiation, which bring faster computation and less memory cost. However, mesh-based representation is not able to simulate highly deformable materials such as fluids. In contrast to them, we opt for a particle-based representation since our work focuses on simulating fluid systems and relevant manipulation skills. We use MPM-based method which naturally supports particle-based materials ranging from solid to fluids, and also the coupling between them.

## F.2 DiSECt

DiSECt (Heiden et al., 2021) is a differentiable tool tailored towards robotic cutting of soft materials. It also opts for a mesh-based representation and FEM to simulate dynamics, which is ideal for computing physical phenomena in cutting scenarios, such as plastic deformation, fracture, and crack propagation. Our work differs from them in that FluidEngine is a general-purpose simulator, and our focus is on manipulation scenarios involving different types of fluids. Therefore, although our simulator does not specialize in special cutting scenarios, it can support a wide range of daily manipulation tasks involving interaction between various material types, ranging from solid, liquid to gas.

## F.3 PhiFlow

PhiFlow (Holl et al., 2020) is a differentiable fluid simulation toolbox. However, the major functionality is a underlying differentiable PDE solver, which can be used to prototype small scenes with fluids but is not capable of building up realisitic 3D robotic manipulation scenarios. It requires manually specifying boundary conditions for fluid simulation, which is not realistic if we want to use actual object meshes for manipulation. Also, it relies on grid-based simulation for fluids, while our particle based simulation is more ideal for simulating liquids, which does not require boundary specifying and supports sparse computation. Moreover, PhiFlow is a special-purpose simulator specific to flow simulation, and doesn't support other material types such as solids, which is important for conducting robotics research in manipulation.

## F.4 JAX-Fluids

Similar to PhiFlow, JAX-Fluids (Bezgin et al., 2022) is a special-purpose differentiable solver tailored to flow simulation. Therefore, it shares the similar limitations of PhiFlow as discussed above: it's not designed for general purpose multi-material simulation, not supporting dynamics of other materials such as rigid and plastic objects, and does not support features such as mesh processing, scene building, and realistic rendering. In contrast, FluidEngine is a full-stack simulation platform providing such features and useful for conducting robotic manipulation research.

## G    DISCUSSION ON TASK SELECTIONS

When designing our tasks, we would like them to cover a wide range of variations, and these are the main factors that we consider: 1) varying material type and their coupling, from inviscid liquid, to viscous liquid, to non-newtonian fluid, to gaseous fluid, 2) the contact between the robot manipulator and the fluid body (direct contact or indirect emitting), and 3) the role of fluids, as a transmitting medium or the manipulation target. Therefore, we came up with the 10 selected tasks. In Table 5, we compare the selected tasks over the above 3 factors, and identify the unique characteristics of each task. We believe our proposed suite of task selections covers a wide range of fluid manipulation tasks encountered in our daily life, and should be useful as a set of standardized tasks for studying fluid manipulation problem.

