# OpenReview forum: "FluidLab: A Differentiable Environment for Benchmarking Complex Fluid Manipulation"
_ICLR.cc/2023/Conference — ICLR 2023 notable top 25%_

### Official Review · Reviewer_fmMN · 2022-10-19

**Confidence:** 5
**Correctness:** 4
**Technical Novelty And Significance:** 4
**Empirical Novelty And Significance:** 4
**Recommendation:** 10

**Clarity, Quality, Novelty And Reproducibility:**

The paper is clear and well written. While it is not novel in the algorithmic sense, it does add a new tool for researchers to use as well as a set of standardized experiments to compare against. Furthermore, because this paper both makes the engine and the experiments public (or will do so upon acceptance according to the paper), reproducibility is high. It should be easy for future researchers to download the code and rerun the experiments on their own.

There are a few minor things missing however. The paper doesn’t mention what the size of each timestep is (e.g., 0.1s). There also isn’t any mention of how fast the engine is for each of the tasks. Does it run in real time on a modern computer? Slower? Faster? Finally, several RL policies are trained on the tasks. It’s never stated what the input is, although it is explicitly stated the input is not rendered images. If the state of the world is represented mostly as particles, it is unclear how those particles are input into the RL policies. This is non-trivial, and should be clarified.


**Strength And Weaknesses:**

This is a strong experimental paper. It presents a differentiable physics engine alongside a series of tasks to demonstrate its value. The engine combines many different techniques to allow it to simulate a variety of material types such as rigid objects, plastics, liquids, and gasses. It also can compute the gradients of the state of the simulation, making it valuable for gradient-based techniques. Furthermore, the paper adds to this a set of experimental tasks designed to show the value of a differentiable physics engine. The tasks span a wide variety, from latte art to ice cream to HVAC control. The results in the paper clearly show that the gradient-based techniques are better able to solve the tasks, and in some cases the gradient-free techniques can’t solve them at all.

The biggest weakness of this paper is that it doesn’t present any novel methodology. It’s essentially an announcement of a new library (FluidEngine) and an empirical argument around why it’s useful (FluidLab). I actually don’t see this as much of a weakness, if at all. With the fast pace of research these days, there is a strong incentive to focus on novel methods and sacrifice all else. This means that things like infrastructure and datasets (everyone makes their own, the days of Imagenet are over) and repeatable experiments often get overlooked. The fact that this paper stops to focus on that is actually beneficial to the field, and can provide a good testbed for future algorithms developed in this area. It’s worth a publication to bring this to the attention of a wider audience.


**Summary Of The Paper:**

This paper presents FluidEngine and FluidLab. FluidEngine is a differentiable physics engine that can simulate rigid objects, deformable objects, liquids, and gasses by combining different methodologies together. FluidLab is a suite of MDP environments for evaluating the efficacy of different algorithms on physical manipulation tasks. The tasks in this paper are diverse (from latte pouring to HVAC control) and manipulate many different kinds of objects. The results show that algorithms that can take advantage of the gradients provided by the engine perform significantly better on the tasks than algorithms that can’t.

**Summary Of The Review:**

Overall, this is a good paper. It presents a new differentiable physics engine and a set of standardized tasks to go with it. The results in the paper clearly show that the use of gradients vastly improves the performance of the policies. While this paper doesn’t present a novel algorithm, its contribution as a platform and experimental paper make it a high quality publication.

---

> ### Author Response · Authors · 2022-11-17
> **Response to Reviewer fmMN**
>
> Thank you so much for your detailed feedback! We are very encouraged that you liked our work and found it a strong experimental paper. We hope to address your concerns below.
>
> ### 1. Algorithmic novelty
> > While it is not novel in the algorithmic sense, it does add a new tool for researchers to use as well as a set of standardized experiments to compare against.
>
> Indeed, we agree that our trajectory optimization method falls under the existing paradigm of optimization using differentiable physics, and is not a completely novel method. However, as we show in our experiments, direct adoption of gradient-based optimization (DP-H) does not work well on our proposed tasks due to the complex optimization problem associated with fluid systems. To address this challenge, we proposed several optimization techniques that are applicable to the context of fluid manipulation, which blend in naturally into the gradient-based optimization framework and are proved to be useful when optimizing for the tasks. We believe such techniques are novel and useful in the domain of fluid manipulation. That said, we are still very glad that you recognize the contribution and novelty of our proposed tasks and simulation platform!
>
> ### 2. Code release
> > Furthermore, because this paper both makes the engine and the experiments public (or will do so upon acceptance according to the paper), reproducibility is high.
>
> Yes, open-sourcing our code is important for assessing and reproducing our work. We have released our code anonymously here: https://github.com/FluidLabICLR2023/FluidLab.
>
> ### 3. Simulation and method details
> > There are a few minor things missing however. The paper doesn’t mention what the size of each timestep is (e.g., 0.1s). There also isn’t any mention of how fast the engine is for each of the tasks. Does it run in real time on a modern computer? Slower? Faster? Finally, several RL policies are trained on the tasks. It’s never stated what the input is, although it is explicitly stated the input is not rendered images.
>
>
> Thanks for bringing up these concerns regarding the details! All our envs use a simulation step of 2e-3 seconds, each containing 10 substeps of 1e-4 seconds for ensuring simulation stability. All our proposed tasks contain around 100,000 particles and run in real time (60 FPS where each frame is a simulation step) on a desktop computer equipped with an Nvidia RTX 3080 GPU and a Intel i7-8700K CPU, and the typical GPU usage is under 50%.
>
> Our simulation engine performs simulation using a consistent data format throughout the full simulation pipeline. All our tasks use FP32 (single-precision floating-point) for fast simulation, which alreay provides decent simulation stability and gradients in our experiments. Our simulation engine also provides the capability of running with FP64, which can be enabled with a single line of code in the configuration file. This would slow the computation but provides more accurate simulation, and we believe this could be useful for other more complex environments that people would potentially build in the future.
>
>
> For RL policies, we use the widely-used public codebase [Stable Baselines3](https://github.com/DLR-RM/stable-baselines3), where the RL policies use the default MLP networks. We downsample the environment state to an ordered subset of particles and cells, and then concatenate the properties of them into a state matrix. Then, this matrix is flattened and fed as input to the policy network.
>
> We have updated our manuscript with the above details in Appendix D.1. Thank you again for your constructive feedback and suggesting these missing details! They are really helpful for improving the quality of our submission.

---

### Official Review · Reviewer_KJH8 · 2022-10-23

**Confidence:** 4
**Correctness:** 4
**Technical Novelty And Significance:** 3
**Empirical Novelty And Significance:** 3
**Recommendation:** 8

**Clarity, Quality, Novelty And Reproducibility:**

The paper is clearly written, well-organized, and should be understandable by people without a background in physical simulations. It refers to prior works as necessary. The authors promise to release the code for the engine and the sample simulations, which should allow users to reproduce the results presented in the paper. The paper itself of course does not offer sufficient space to fully document a complex engineering artifact like FluidEngine.

Additional novelties introduced in the paper are the set of complex tasks forming FluidLab, and a couple of
techniques for making better use of gradients and stabilizing the optimization problem (soft contact model, starting with a restricted temporal region, cross-particle gradient sharing).

Questions/suggestions:
* What are the limitations of the solver in terms of density ratios, Reynolds numbers, etc? Are there any concerns about numerical stability?
* How is mass taken into account in the SDF representation?
* The text mentions that the mesh provides boundary conditions for the fluid; is momentum transfer from the fluid to the mesh allowed as well, i.e. is the coupling bidirectional?
* I found it surprising that gases are simulated as an *incompressible* fluid, when one of the basic properties of gases is their compressibility. Could you please clarify the methodological choices here?


**Strength And Weaknesses:**

The paper is built around FluidEngine, which is also one of its main strengths. It represents a significant aggregation of capabilities in a single simulator, with alternatives falling short in terms of supported types of matter, differentiability, and source code access. An additional advantage in this context is the usage of the OpenAI Gym API.

A weak side of this work is the lack of information about the validation of the physics engine. The authors are using existing methods, but implement them from scratch. A physics engine is a complex piece of software, which can suffer from subtle bugs which are hard to notice outside of a rigorous evaluation in setups with known solutions.


**Summary Of The Paper:**

The paper introduces FluidLab, a platform for simulating interactions of complex fluids and solid objects. The platform includes a fully-differentiable and GPU-accelerated physics engine (FluidEngine), as well as an OpenGL renderer. FluidEngine is used to evaluate a few existing algorithms in the context of the FluidLab task set -- a set of 10 complex fluid manipulation tasks modeled on real-world scenarios that the authors define and introduce in the paper. The experimental part of the paper compares model-free RL to trajectory optimization based on sampling and differentiable physics. Access to gradient information of the simulation is shown to significantly benefit trajectory optimization.


**Summary Of The Review:**

Overall, this appears to be a significant piece of work, introducing a very useful piece of software, and proposing a set of standard tasks which require more complex physical interactions than those typically used in similar setups. The authors correctly note that a lot of prior work in the field of fluid manipulation involves relatively simple scenarios. The paper represents a significant advance towards more complex settings.

---

> ### Author Response · Authors · 2022-11-17
> **Response to Reviewer KJH8 (Part 1)**
>
> Thanks a lot for your constructive and detailed feedbacks! We are very grateful that you recognize our proposed tasks and simulation engine as a significant advance towards more complex settings. Below we address your concerns.
>
> ### 1. Validation of the physics engine
>
> > A weak side of this work is the lack of information about the validation of the physics engine. The authors are using existing methods, but implement them from scratch. A physics engine is a complex piece of software, which can suffer from subtle bugs which are hard to notice outside of a rigorous evaluation in setups with known solutions.
>
> Thank you very much for pointing out the helpfulness of certain validations of our physics engine! We implemented the physics engine from scratch because our proposed tasks requires simulating a wide range of matters and their interactions, which is not supported by any of the existing platforms. We definitely agree that such a peice of software could potentially suffer from subtle bugs and certain validations are indeed very important to make our work more reliable and useful.
> In order to address your concern, we made the following efforts:
> 1. One useful way of validating the correctness of simulation engines is to run simulation experiments and verify if the produced simulation results match known physical phenomena. Below we list a number of classical physical phenomena that are typically observable in rigid body and computational fluid dynamics. We testified our simulation engine by running simulations to see if it can successfully reproduce the listed behaviors. We have updated our [project website here](https://sites.google.com/view/fluidlab/fluidengines-validation-exps) with visual results of these validation experiments. We believe running against such validating scenarios could help demonstrate the correctness and reliability of our simulation engine, and reveal potential issues.
>     - **Kármán vortex street**: When a flow of fluid is separated by blunt bodies, a repeating pattern of swirling vortices would appear due to vortex shedding. We simulate a flow of fluid flowing by a rigid cylinder and visualize the top-down view. We colorized the fluid around the cylinder, and it can be seen that our simulator can faithfully produce a street of vortices, typically known as the Kármán vortex street phenomenon.
>     - **Magnus effect**: This is an observable phenomenon commonly associated with a spinning object moving through a fluid. The path of the spinning object is deflected in a manner not present when the object is not spinning. One example of such phenomenon seen in daily life is the screw shot in football playing. Here we simulate a spinning ball in a fluid body. It can be observed that the spinning motion produces a deflection in the motion trajectory of the ball, and the extent of the deflection is dependent on its spinning velocity.
>     - **Buoyancy**: We verify here if objects with different density in a liquid body would incur different magnitude of buoyancy. It can be observed that the red all would float on the water surface due to buoyancy.
>     - **Incompressibility and stable volume**: Liquids are generally considered incompressible. Our MPM-based engine simulates weakly-compressible liquids, where the momentum and mass are updated every step via consecutive grid-particle-grid information passes. Therefore, one concern is if the simulated fluid bodies can maintain consistent pressure and volume through long-horizon simulations, without momentum and pressure loss. We verified our simulator can simulate a stable fluid body that maintains steady volume and pressure after 10,000 simulation steps.
>     - **Conservation of momentum**: When objects are in collision, their motions obey conservation of momentum. Here we test the collision of two objects with exactly the same mass on a frictionless floor, where it can be observed that they exchanged their momentum after the collision as expected.
>     - **Varying bouncing behaviors with varying plasticity**: Here we show different behaviors of the object bouncing on the floor when we vary the plasticity of the object. From left to right, we increase the plasticity of the object. Our simulator can simulate bouncing behavior with varying plasticity, where it can be observed energy dissipates faster with a greater plasticity.

---

> > ### Author Response · Authors · 2022-11-17
> > **Response to Reviewer KJH8 (Part 2)**
> >
> > ### 1. Validation of the physics engine (continued)
> > 1. (continued)
> >     - **Rayleigh–Taylor instability**: This phenomenon describes the instability of an interface between two fluids with different densities, which occurs when the heavy component is above and pushing the light one. Here we simulate the behavior of two layers of fluids with different densities, where it can be observed that a plume of heavy fluid with small vortices emerges and evolves due to the interfacial instabilities.
> >     - **Dam break**: This is a classical engineering test case, where a fluid volume falls due to gravity and splashes within the domain. We successfully reproduced this phenomenon with our simulation engine.
> >
> >     Our simulators successfully reproduced these physical phenomena, as we show on our project site. We believe running such validation experiments could informatively help demonstrate the accuracy and reliability of our simulation engine.
> >
> > 2. Our work proposes a set of manipulation tasks motivated by real-world scenarious, so the reliability and usefulness of our simulation platform could be demonstrated and validated if the optimized policies can be used in real-world robotic experiments. We have added experiments in real-world, where we apply optimized trajectories from simulation in real world and qualitatively evaluate if such optimized trajectories can accomplish the designed tasks. We conducted four experiments proposed in the paper on our real robot setup: Latte Art (Stirring), Ice Cream (Dynamic), Mixing and Gathering. We have updated our manuscript and project site with such real-world results **[here](https://sites.google.com/view/fluidlab/real-world-results)**.  It can be seen that although there's definitely certain sim-to-real gap in terms of material behavior and dynamics, when we apply the simulation-optimized trajectories to real-world, the tasks can be completed to a reasonable extent. We believe this suggests that our developed simulation engine is useful and reliable, and we hope this could also help address your concern regarding the validity of our engine.
> > 3. We have released our code anonymously here!: https://github.com/FluidLabICLR2023/FluidLab We invite researchers from the open research community to utilize and test our simulation platform, and we will keep monitoring our codebase and merging pull requests. Our goal is to provide an easy-to-use platform for researchers to both evaluate their methods in fluid manipulation settings, as well as to collaboratively build up more challenging yet useful manipulation scenarios using our simulation engine. We believe releasing our source code publicly could definitely be helpful in revealing potential issues (if there's any) and make the platform more reliable.
> >
> > ### 2. Questions/suggestions
> > > What are the limitations of the solver in terms of density ratios, Reynolds numbers, etc? Are there any concerns about numerical stability?
> >
> >
> > Indeed, being numerically stable is an important aspect of a simulation engine. In Table 4 in the appendix, we listed the material properties used in our experiments. Based on our numerical experiments, our simulator is stable within a density ratio of 1:50, which is wide enough to cover most of the multi-component fluid simulations (e.g., coffee mixing) in daily settings. Note that our solver currently cannot handle multi-phase fluid tasks with contrasting density ratios such as bubble flow, which is out of scope of our current work. Based on our numreical tests, our simulator supports a Reynolds number up to around 10^3. For the grid-based simulator (e.g., smoke scenes), our solver does not have any stability concerns because of the unconditionally stable semi-Lagrangian advection schemes we employed. For the scenes involving particles, our solver uses Symplectic Euler integration and it has a CFL < 1 restriction to ensure its numerical stability. Our current experiments use a simulation substep of 2e-4 seconds across all the tasks, which shows to be stable enough for the material properties adopted in our tasks. In case of presence of unstability, increasing the temporal resolution to e.g. 1e-4 seconds per substep could bring back the simulation to a stable region again. In short, we did not observe any stability issue with our current material and task selections, and such stability can be further enhanced by increasing temporal resolution of the simulator in case of instability (at the cost of reduced simulation speed) to ensure the CFL condition is satisfied.

---

> > > ### Author Response · Authors · 2022-11-17
> > > **Response to Reviewer KJH8 (Part 3)**
> > >
> > >
> > > > How is mass taken into account in the SDF representation?
> > >
> > > Our simulator supports bi-directional coupling. For the SDF-based rigid objects, the engine computes forces and wrenches (w.r.t the the CoM of the object) exerted by all the particles, and then computes the translational and angular acceleration of the objects accordingly before updating their states.
> > >
> > > > The text mentions that the mesh provides boundary conditions for the fluid; is momentum transfer from the fluid to the mesh allowed as well, i.e. is the coupling bidirectional?
> > >
> > > Yes, that is allowed. Meshes are treated as boundary conditions for the grid-based gas simulation. The momentum transfer is computed in the same way as described in the above question, where the forces are computed based on the momentum change of each grid of the gas field, and then applied on the corresponding contact point of the boundary mesh. Note that in our proposed tasks, this effect is not included since our tasks only consider static boundarys for the gas field, but this can be supported by our engine.
> > >
> > > > I found it surprising that gases are simulated as an incompressible fluid, when one of the basic properties of gases is their compressibility. Could you please clarify the methodological choices here?
> > >
> > > In a low-speed (below sound speed) and single-phase (without bubbles) setting, gas is nearly incompressible and can be simulated using an incompressible Navier-Stokes solver. This has been a common practice in computer graphics to animate smoke. Compressible gases are only used when there are shock waves (e.g., explosion) in the scene, which is out of the scope of our current work and task scenarios.
> > >
> > > Thank you again for your helpful suggestions and concerns. They are really useful for improving the quality of our submission! We have updated both our manuscript and our project site to reflect the additional experiments and results.

---

> > > > ### Comment · Reviewer_KJH8 · 2022-11-27
> > > > **Thanks**
> > > >
> > > > Thank you for the detailed response, and for adding the various validation tests. I'm glad to see that all the test cases check out!

---

### Official Review · Reviewer_3F7A · 2022-10-24

**Confidence:** 4
**Correctness:** 2
**Technical Novelty And Significance:** 2
**Empirical Novelty And Significance:** 3
**Recommendation:** 6

**Clarity, Quality, Novelty And Reproducibility:**

As mentioned above, it is an interesting direction, but the advantages over existing simulators are left unclear in the current submission. A better motivation (and potential adjustment) of the test cases would also be a good idea.


**Strength And Weaknesses:**

The paper presents an interesting range of test cases, and fluid interactions are definitely and interesting and challenging task for learning methods. The test cases also exhibit a nice complexity, and the paper contains a good range of classic RL methods that are compared to the differentiable physics variant.

A big question that I had while reading is also how this engine is positioned in the existing environment of differentiable simulators. For the MPM cases, the engine seems to build on Taichi, but doesnt make clear how it extends it. Taichi already provides gradients since DiffTaichi - so is it merely a selection of scenes for Taichi, or are there changes to the simulator? The paper is currently written to advertise the "FluidEngine" simulator. However, it does not argue why it wold make sense to choose this simulator over existing ones, i.e. Taichi for MPM, or Phiflow for single phase cases (a discussion of the latter seems to be missing). In order to claim this system as a contribution, I think it would be important to run comparisons to show what advantages are to be gained over alternatives such as Taichi, Phiflow etc.

The selection of test cases also seems somewhat arbitrary to me. Why is there such a large focus on latte art and ice cream? These cases make for nice visuals, but if the goal of the paper is to establish these scenes as a more widely used benchmark, this seems strange to me. The remaining 6 cases establish a wider range of different behavior, and look more meaningful to me. In general, I would recommend to add structure to the selection and ordering of the cases. Also it would be good to argue why these particular scenes are worth considering as test cases. I definitely found it strange to start with "latte-art" instead of basic settings than build up to more complex ones.

A smaller note, but I would also caution against statements that the DP variants yield fundamentally better learned states than the RL variants. In the limit of infinite compute a well-tuned RL algorithm will eventually discover these as well. DP might get there much faster, but there's no reason RL should not discover these states at some point.

**Summary Of The Paper:**

This paper describes a differentiable simulator for a series of tasks tailored towards robotic interactions. The system is MPM-based, and presents a range of different tasks that focus on interactions with fluids, all the way from two-dimensional transport to 3D manipulations. The paper then reiterates on the argumentation from previous work that using directional information in the form of gradients outperforms model-free RL approaches. On the side, the paper also briefly discusses modifications of the simulation based gradient calculations.

**Summary Of The Review:**

My main open question is that of advantages over existing simulators. Table 1 is not sufficient here: other differentiable fluid simulators like (Diff)Taichi, Phiflow and JAX-Fluids are already out there, so I don't understand why they're left out, and how the work distinguishes itself here. The RL comparisons are interesting, and the tweaks for proper convergence likewise. Here, I think the paper undersells a bit: there's likewise no effort to put this into context of existing works at ICLR, and as such they don't play a very prominent role. Thus, taken together from the current state of the submission, this looks like a paper below the acceptance threshold to me.

---

Post rebuttal: the authors have provided a nice update, together with the other assessments, I'd be happy to support accepting this paper.

---

> ### Author Response · Authors · 2022-11-17
> **Response to Reviewer 3F7A (Part 1)**
>
> We really appreciate your detailed comments and criticisms! We are very encouraged that you found our proposed tasks intersting and complex. We believe studying these tasks is an important step towards more intelligent robotic agent. Your concerns regarding the role and position of FluidEngine and our task selection are very reasonable, and we hope to address your concerns in our response below.
>
> ### 1. Comparison with other environments
> > A big question that I had while reading is also how this engine is positioned in the existing environment of differentiable simulators. For the MPM cases, the engine seems to build on Taichi, but doesnt make clear how it extends it. Taichi already provides gradients since DiffTaichi - so is it merely a selection of scenes for Taichi, or are there changes to the simulator? The paper is currently written to advertise the "FluidEngine" simulator. However, it does not argue why it wold make sense to choose this simulator over existing ones, i.e. Taichi for MPM, or Phiflow for single phase cases (a discussion of the latter seems to be missing). In order to claim this system as a contribution, I think it would be important to run comparisons to show what advantages are to be gained over alternatives such as Taichi, Phiflow etc.
> > My main open question is that of advantages over existing simulators. Table 1 is not sufficient here: other differentiable fluid simulators like (Diff)Taichi, Phiflow and JAX-Fluids are already out there, so I don't understand why they're left out, and how the work distinguishes itself here.
>
> We apologize for causing such confusions and not explaining detailed differences with related works in our original submission. We discuss the differences in detail here.
> - **Taichi** First, we would like to make some clarifications regarding Taichi. Taichi by itself is just a domain-specific programming language that provides GPU-accelerated parallel computation, and we use it to implement FluidEngine and FluidLab. DiffTaichi is the automatic differentiation system of Taichi that supports gradient computation of numerical operations coded using Taichi. Essentailly, Taichi and DiffTaichi are like PyTorch and it accompanying Autograd feature. On the other hand, Taichi and DiffTaichi do come with many example codes of simulating different scenarios, but they are scene-specific simulation examples and there's a big gap between them and an actual simulation environment that can be used for repeatable robotics research. We list a number of major differences here. 1) DiffTaichi doesn't support building computation graph and gradient checkpointing, which is necessary for computing and storing gradient flow over long-horizon manipulation tasks, especially when GPU memory is a major bottleneck. Therefore, unlike in PyTorch, any computation step involving gradient flow needs manually setting up the computation graph, and our framework handles that. 2) Taichi does provide MPM-based simulation examples, but they don't support functions such as mesh loading, SDF computation, SDF-based collision detection, friction modelling, mesh to particle conversion, etc. 3) Taichi doesn't support grid-based 3D smoke simulation, and doesn't support coupling between smoke, liquid, and mesh-based rigid objects, which is precisely one of the contributions of our platform, since such coupling is necessary for studying realistic fluid manipulation problem. 4) DiffTaichi only provides gradient computation of simple operations, not supporting gradients for e.g. SVD computations and the projection step in gas field simulation. 5) Our proposed FluidEngine provides a unified configuration system and interface for building custom environments and specifying robotic agents, and also provides function for storing and accessing gradients of the intermediate states and robot actions. 6) The examples provided with Taichi typically requires saving scene configurations to a file and using an external rendering software such as Houdini for visualization, while we provides a stand-alone OpenGL-based renderer integrated in FluidEngine, running in real time and providing easy visualization and also capability for developing image-based policy learning method. 7) FluidLab is a set of standardized manipulation tasks for developing and evaluating robot learning algorithms, built on top of FluidEngine. This is like SoftGym versus FleX.
> In summary, FluidEngine is a piece of complete software with various APIs that can be used easily for building up robotic research environments, just like what people are able to do with PyBullet or Mujoco, except that FluidEngine additionally supports deformable and fluid materials, while also providing gradient information.

---

> > ### Author Response · Authors · 2022-11-17
> > **Response to Reviewer 3F7A (Part 2)**
> >
> > (Continued)
> > - **PhiFlow** PhiFlow is indeed a very relevant simulation toolbox. However, the major functionality is an underlying differentiable PDE solver, which can be used to prototype small scenes with fluids but is not capable of building up realisitc 3D robotic manipulation scenarios. It requires manually specifying boundary conditions for fluid simulation, which is not realistic if we want to use actual object meshes for manipulation. Also, it relies on grid-based simulation for fluids, while our particle based simulation is more ideal for simulating liquids, which does not require boundary specifying and supports sparse computation. Moreover, PhiFlow is a special-purpose simulator specific to flow simulation, and doesn't support other material types such as solids, which is important for conducting robotics research in manipulation.
> > - **JAX-Fluids** Similar to PhiFlow, JAX-Fluids is a special-purpose differentiable solver tailored to flow simulation. Therefore, it shares the similar limitations of PhiFlow as discussed above: it's not designed for general purpose multi-material simulation, not supporting dynamics of other materials such as rigid and plastic objects, and does not support features such as mesh processing, scene building, and realistic rendering. In contrast, FluidEngine is a full-stack simulation platform providing such features and useful for conducting robotic manipulation research.
> >
> > We originally designed Table 1 to include comparisons between FluidEngine/Lab and existing popular simulation platforms that are used for robotics and manipulation research. However, we definitely agree that a more clear comparison with Taichi, PhiFlow and JAX-Fluids would help better position our work in the literature, and we would like to thank the reviewer again for this valuable suggestion. We have updated Table 1 in our manuscript to include Phiflow and JAX-Fluids; in addition, in Appendix G, we added detailed comparisons in text with Taichi, Phiflow, JAX-Fluids, as well as other differentiable simulation tools like SPNets, DiffSim and Disect. We hope such additional comparisons could address your concerns regarding the difference between FluidEngine and other differentiable simulators.

---

> > > ### Author Response · Authors · 2022-11-17
> > > **Response to Reviewer 3F7A (Part 3)**
> > >
> > >
> > > ### 2. Rationale behind our task selections
> > >
> > > > The selection of test cases also seems somewhat arbitrary to me. Why is there such a large focus on latte art and ice cream? These cases make for nice visuals, but if the goal of the paper is to establish these scenes as a more widely used benchmark, this seems strange to me. The remaining 6 cases establish a wider range of different behavior, and look more meaningful to me. In general, I would recommend to add structure to the selection and ordering of the cases. Also it would be good to argue why these particular scenes are worth considering as test cases. I definitely found it strange to start with "latte-art" instead of basic settings than build up to more complex ones.
> > >
> > > Thanks for bringing this up! This is a very valuable point that we missed in paper writing, and we agree that making our task selections more structured will make the paper better motivated.
> > >
> > > The reason that we start with latte art is that this specific scenario is the very original motivation of the paper. Today, there's already robobarista capable of helping people make coffee, but they are still far from accomplishing more complex tasks such as making goal-conditioned latte art. Thinking about this, we realized that many daily manipulation tasks involve various forms of fluids -- which poses major challenges on today's robot learning methods due to the complex state space and the underlying dynamics. However, we agree that from the paper writing perspective, we should have ordered our task selection in a more structured way.
> > >
> > > When designing our tasks, we would like them to cover a wide range of variations, and these are the main factors that we consider: 1) varying material type and their coupling, from inviscid liquid, to viscous liquid, to non-newtonian fluid, to gaseous fluid, 2）the contact between the robot manipulator and the fluid body (direct contact or indirect emitting)， 3）the role of fluids, as a transmitting medium or the manipulation target. Therefore, we came up with the 10 proposed tasks, which we believe covers enough variations along these factors, and should be representative for studying fluid manipulation problems encountered in our daily life.
> > > We included a discussion on our rationale in Appendix H, and presented a detailed comparison of these tasks and their variations along the above 3 factors in Table 5. We additionally included the unique characteristics of each task in the table. We hope the added discussion and table could explain better the rationale behind our task selections.
> > >
> > > In addition, per your request, we revised the ordering of our tasks in both Figure 2 and Section 4.1. Now the ordering is more logical following this stucture: We started with relatively simple tasks with only inviscid liquid (Pouring). Then we move to tasks with coupling between liquid and rigid objects, where the liquid is used as a transmitting medium (Gathering and Transporting). Afterwards, we move to tasks with coupling between multiple fluid bodies: first is Mixing, where there's no a fine-grained goal configuration, then the two latte art tasks, one with the a relatively simple pouring setting, and the second one with a more complex coupling between multiple fluid bodies and the rigid stirrer, both requiring more fine-grained goal configurations. Now we are done with Newtonian fluids, and we move to tasks with more complex non-Newtonian fluids, i.e. the ice cream tasks, with a increasing difficulty, since the the first one is relatively static, and the second one involves dynamic interactions between the rigid manipulator and the ice cream. Finally, we introduce the Air Circulation task, which involves the highly dynamic motion of gaseous air flows.
> > >
> > > We hope the revised ordering of tasks selections is more reasonable! It follows variation of the above 3 factors, with an increasing complexity.
> > >
> > > > A smaller note, but I would also caution against statements that the DP variants yield fundamentally better learned states than the RL variants. In the limit of infinite compute a well-tuned RL algorithm will eventually discover these as well. DP might get there much faster, but there's no reason RL should not discover these states at some point.
> > >
> > > Yes, that is absolutely correct. We have revised our discussion in the RL paragraph with additional terms like "within in limited training iterations" to make it more accurate.
> > >
> > > ### 3. Additional real-world experiments
> > > We have also updated our project project site with [visual results here](https://sites.google.com/view/fluidlab/real-world-results) with additional real-world experiments, and we hope this could help better illustrate the value of our work.
> > >
> > >
> > > We would like to thank you again for such a detailed feedback! It definitely helped significantly improve the quality of our paper. We hope the revised manuscript now presents and explains our contributions better.

---

> > > > ### Comment · Reviewer_3F7A · 2022-11-25
> > > > **Rebuttal**
> > > >
> > > > The authors have submitted a nice update of their work that addresses most concerns I had regarding the presentation. The paper now makes clearer why another "layer" build on top of Taichi is necessary. So while I'm still a bit surprised by the very high score of a fellow reviewer, overall I'd be happy to recommend accepting this paper. I've raised my score to indicate this.
> > > >
> > > > I still have one request for a "final" version, though: the appendix G.4 now discusses in more detail where the shortcomings of Taichi and DiffTaichi lie. As these are not obvious from the available papers, please include more of this discussion early on in the main paper. Otherwise readers will wonder while reading all the way until G.4 about this point.

---

> > > > > ### Author Response · Authors · 2022-11-25
> > > > > **Response to reviewer**
> > > > >
> > > > > Thank you very much! We will follow your suggestion and add such description in the main paper of our final revision. Meanwhile, we were wondering if it's possible to revise the correctness score since we believe there shouldn't be any correctness issue left in our updated paper.
> > > > >
> > > > > Thanks a lot for your reply again!
> > > > > Best, authors.

---

### Official Review · Reviewer_Frwu · 2022-10-25

**Confidence:** 4
**Correctness:** 3
**Technical Novelty And Significance:** 3
**Empirical Novelty And Significance:** 3
**Recommendation:** 6

**Clarity, Quality, Novelty And Reproducibility:**

Clarity: the paper is generally clear and easy to read.
Quality: in general, the paper is of decent quality.
Reproducibility: considering the system is complex, without the code, it is hard for one to judge the reproducibility.

**Strength And Weaknesses:**

Strength:

1. This work covers a variety of interesting fluid manipulation tasks, such as ice cream and latte art, while being fully differentiable.

2. This work also supports complex fluidic material behavior.

3. It supports customization of the environment.


Weaknesses:

1. The sim-to-real gap can be a significant concern. Since there is no real robot execution, the reader does not know how much of a gap there is between the proposed simulation and the real world.

2. The proposed benchmarking effort does not benchmark SOTA methods for fluid manipulation. The benchmark itself should measure the performance of recent SOTA methods to show real progress in this area of research and to provide a promising direction. Methods of fluid manipulation can be considered, for example, SPNets [1].
Methods of differentiable RL can be considered: SHAC [2], PODS [3].


3. Table 1 compares existing simulators with rigid and liquid object types, with some important works left out of the literature review. These works are also differentiable and support various object types: DiffSim [4], Disect [5].


4. Key insights from the benchmark results are missing. Considering these results, it would be more valuable to provide a clear direction or insight to guide the researcher.

[1] Connor Schenck, Dieter Fox. Spnets: Differentiable fluid dynamics for deep neural networks. CoRL, 2018.

[2] Jie Xu, Miles Macklin, Viktor Makoviychuk, Yashraj Narang, Animesh Garg, Fabio Ramos, and Wojciech Matusik. Accelerated policy learning with parallel differentiable simulation. ICLR, 2022.

[3] Miguel Zamora, Momchil Peychev, Sehoon Ha, Martin Vechev, Stelian Coros. PODS: Policy Optimization via Differentiable Simulation. ICML, 2021.

[4] Yi-Ling Qiao, Junbang Liang, Vladlen Koltun, and Ming C. Lin. Scalable differentiable physics for learning and control. 2020.

[5] Eric Heiden, Miles Macklin, Yashraj Narang, Dieter Fox, Animesh Garg, and Fabio Ramos. Di-sect: A differentiable simulation engine for autonomous robotic cutting. RSS, 2021.

Questions:

1. How helpful is gradient sharing? Are there any ablation study results to help with the understanding?

2. The task of Latte Art seems to be highly nonconvex, but DP and DP-H accomplish this task almost perfectly. We all know that differential-based trajectory optimization eventually reaches a local minimum, how can DP and DP-H tackle the local minimum problem and reach the global optimum? In detail, what are the initial actions of DP and DP-H?

**Summary Of The Paper:**

This paper presents FluidLab, a simulation environment with a diverse set of manipulation tasks involving fluid. The simulation environment is supported by FluidEngine, which is built with Taichi. In addition, the paper also provides some domain-specific optimization techniques to best utilize differentiable physics in liquid manipulation.


**Summary Of The Review:**

I believe validating the simulator with the real robot is a must when the task is designed for robot manipulation. In addition, as a benchmark paper, demonstrating recent SOTA methods to show real progress in this area of research and providing a promising direction is very important.

---

> ### Author Response · Authors · 2022-11-17
> **Response to Reviewer Frwu (Part 1)**
>
> Thank you for your detailed comments and suggestions! We really appreciate that you found our proposed tasks interesting and you recoginize the value of our simulator. We have added additional experiments and results per your requests, and we hope to address all your concerns below.
>
> ###  1. Real robot execution to validate the simulator
> > The sim-to-real gap can be a significant concern. Since there is no real robot execution, the reader does not know how much of a gap there is between the proposed simulation and the real world.
> > I believe validating the simulator with the real robot is a must when the task is designed for robot manipulation.
>
> - That is a very valid point! Indeed, although we believe our simulation platform provides an efficient test bed and useful gradient information for developing algorithms to deal with manipulation problems involving such a wide range of materials (which is not supported by any existing simulation engines), testing how the optimized policies perform in a real-world setting would be very helpful and informative in evaluating the sim-to-real gap and suggesting future research directions.
> - Per your request, we conducted experiments on a real robot system, with a 7-DoF Franka Emika robot equipped with a parallel jaw gripper. We picked four representative tasks proposed in FluidLab, including **Latte Art (Stirring)**, **Ice Cream (Dynamic)**, **Gathering** and **Mixing**, and set up corresponding real world scenarios to evaluate our method and simulation. These 4 tasks cover both Newtonian and non-Newtonian fluids, as well as interactions between fluids and non-fluid bodies. We have updated our project site with **[visual results here](https://sites.google.com/view/fluidlab/real-world-results)**.
> - We optimize the trajectories in simulation, and then command the robot to execute them using velocity control in the corresponding real-world scenarios. It can be observed that although there’s certain sim-to-real gaps in terms of material behavior and dynamics, when we apply the simulation-optimized trajectories to real-world, the tasks can be completed to a reasonable extent. The sim-to-real gap mainly comes from the simulation inaccuracy and difference in material properties between sim and real. For example, in the latte art experiment, the frothed milk behaves a bit differently than in simulation: it's more sticky and tend to mix again after the latte art pen passes through. This could be further improved in simulation by developing more accurate material models, incorporating feedback from the real world observation, as well as increasing the simulation resolution and number of particles, which we will strive to improve in our future work. Another example is the ice cream experiment, where it is difficult to maintain a steady flow speed of the cream, resulting in a slightly different end shape compared to the simulated one.
> - Although there exists such sim-to-real gap in material dynamics, the robot is able to complete the proposed tasks reasonably well. We would also like to acknowledge that given our current resources, we are not able to conduct all the proposed tasks in real world, such as the air circulation task, which remains as our future work. However, We believe the current 4 selected tasks cover a representative range of materials and task settings, and such experiments could shed light on the value of our simulator as a test bed for robotic research, point out potential sim-to-real gap and suggest possible improvements in our future work.
> - In addition, another way of validating a simulation engine is to run controlled simulations and test if it can reproduce known physical phenomena observable in real-world scenarios. We ran 8 controlled simulation scenarios for a number physical phenomena that are typically observable in rigid body and fluid dynamics, including: **Kármán vortex street**, **Magnus effect**, **buoyancy effect**, **stable volume of liquid**, **conservation of momentum**, **bouncing behaviors**, **Rayleigh–Taylor instability**, and **dam break**. We have included visual results of these simulations in our [project website here](https://sites.google.com/view/fluidlab/fluidengines-validation-exps). Our simulation engine is able to faithfully reproduce the expected physical phenomena. We believe such validations could further help demonstrate the accuracy and reliability of our simulation engine.
>
> We have also revised our manuscript to include both real-robot experiments and the validation simulations.

---

> > ### Author Response · Authors · 2022-11-17
> > **Response to Reviewer Frwu (Part 2)**
> >
> >
> > ### 2. Comparison with other methods
> > > The proposed benchmarking effort does not benchmark SOTA methods for fluid manipulation. The benchmark itself should measure the performance of recent SOTA methods to show real progress in this area of research and to provide a promising direction. Methods of fluid manipulation can be considered, for example, SPNets [1]. Methods of differentiable RL can be considered: SHAC [2], PODS [3].
> >
> > Thank you very much for pointing out this! We very much agree such comparison would help illustrating the real challenges of the proposed tasks.
> >
> > - **Compairson with SPNets** SPNets is certainly an important prior work in robotic manipulation of fluids. We discussed our difference to SPNets in our related work section. However, comparing our method with SPNets on the proposed set of tasks is not very informative due to the following reasons. SPNets is essentially a fluid simulator implemented as neural network layers, which makes it differentiable. Therefore, it is more like a differentiable simulator itself rather than a policy learning method. The parameters of SPNets are the physical parameters of the fluids. Also, one big limitation of SPNets is that its simulation capability: it only supports simulating single-phase non-Newtonian fluids, and only supports one-way coupling from static boundaries to the fluids, i.e. it doesn't support computing effects of fluids on other objects. In fact, this is precisely one of the key contributions and motivations of our simulation framework: we realize that in order to study realistic fluid manipulation problems that are motivated by real-world scenarios with realistic complexity, it's crucial to be able to support a wide range of materials and their inter-material couplings. Our motivation is to study problems beyond the single-phase water tasks proposed in SPNets, such as water pouring and catching, where only the effect of manipulators on the water is considered. Therefore, on the simulator side, SPNets is not able to simulate most of our propose tasks. In addition, on the task solving side, in the SPNets paper the author did propose to use gradients provided by the differentiable simulation to optimize trajectories for liquid control. This is similar to our differentiable physics based optimization method (DP-H), except that we proposed a few optimization techniques (DP) to help solving the tasks. That being said, we definitely agree that a clear discussion on comparison with SPNets would be helpful. We have revised our manuscript to include such a discussion in Appendix G.1.
> > - **Comparison with methods of differentiable RL** Yes, we agree that such a comparison is definitely needed. We chose PODS over SHAC to run such a comparison, because SHAC requires parallel simulation environments to run, which is applicable when the task setting has a relatively low dimensional state space and light-computation dynamics simulation cost, such as the tasks used in the SHAC paper. As suggested in the SHAC paper, the method leverages gradient information of short horizons and thus needs at least 32 parallel simulation envs to show reasonable performance. However, this is not practical in our case since fluid simulations are very computationally expensive, and most of our tasks simulate over 100,000 particles. On the other hand, PODS doesn't have such a requirement, becoming a reasonable candidate to run a comparison against. We ran PODS on our proposed tasks and included the updated results in Section 5.2 and Table 2. PODS performs well in relatively convex tasks such as Latte Art (Pouring), and even outperforms DP and DP-H on simple tasks like transporting, mainly due to its exploratory nature from RL. However, it is not able to match performance in other complex tasks, especially in those where DP shows a clear advantage over DP-H and RL methods. That said, we believe it is not really *fair* to claim that our method is better than PODS, because although PODS uses local gradient information without our proposed optimization techniques, it is also additionally learning a closed-loop policy which maps observation to action, where DP is only optimizing a trajectory. In fact, what PODS is learning is precisely the next step of our research, where we believe a better scene representation to describe the fluid system is needed to better distill DP-optimized trajectory to a closed-loop policy network. Therefore, we thank the reviewer for suggesting this comparison, which suggests two future research directions: 1. how to encode the scene of complex fluid systems in a more informative representation for better policy learning, and 2. we expect better performance by combining our proposed optimization landscapes with methods that uses both differentiable simulation and RL, which we save for our future work.

---

> > > ### Author Response · Authors · 2022-11-17
> > > **Response to Reviewer Frwu (Part 3)**
> > >
> > >
> > > ### 3. Comparison with other simulators
> > > > Table 1 compares existing simulators with rigid and liquid object types, with some important works left out of the literature review. These works are also differentiable and support various object types: DiffSim [4], Disect [5].
> > >
> > > Thanks for pointing out these relevant works! We have updated Table 1 with these simulators. In addition, we belive the table alone is insufficient to describe the their differences to FluidEngine, since some of them are powerful tools in specific scenarios (e.g. Disect for cutting); therefore, we added discussions on comparison with these individual simulators in Appendix G to better position FluidEngine in the literature. In short, DiffSim uses a mesh-based representation and implicit differentiation for faster computation. Disect also uses meshes and specializes in accurate simulation of robotic cutting. Fluid simulation is out of scope of these simulators. In contrast to them, our work focuses more on manipulation scenarios involving fluids, which requires modeling interactions between a wide range of materials. Our underlying physics engine is more general purpose and supports a wider range of materials ranging from  solid, liquid to gas.
> > >
> > > > Key insights from the benchmark results are missing. Considering these results, it would be more valuable to provide a clear direction or insight to guide the researcher.
> > >
> > > Thank you this valuable suggestion! We have added a independent paragraph (**Potential future research directions** in Section 5.2) in our manuscript to discuss current challenges and potential research directions that people could work on with FluidLab.
> > >
> > > ### 4. Questions
> > > > How helpful is gradient sharing?
> > >
> > > We found that in majority of the tasks, soft contact and temporally expanding optimization region are the most important techniques for stabilizing the optimization. Gradient sharing is especially useful in tasks where there's this local optimum problem we described in the pouring example. We use this technique in Latte Art (Stirring), Mixing and Pouring. In the previous two, removing gradient sharing would result in a slight performance drop. In the fine-grained pouring task, without such gradient sharing, trajectory optimization cannot produce meaningful result at all. We added a table (Table 6 in Appendix I) to list the optimization techniques used by DP in each task.
> > >
> > > > The task of Latte Art seems to be highly nonconvex, but DP and DP-H accomplish this task almost perfectly. We all know that differential-based trajectory optimization eventually reaches a local minimum, how can DP and DP-H tackle the local minimum problem and reach the global optimum? In detail, what are the initial actions of DP and DP-H?
> > >
> > > As we described in the paper, we use a temporal trajectory of the desired goal pattern for the latte art tasks, and compute Chamfer distance at each step. This is to ensure a dense reward and to have a fair comparison between RL methods. Such dense goal information and reward makes the optimization problem a lot easier. In our experiments, the initial actions of DP and DP-H are simply zero actions. As we also mentioned, a more ideal task setting would be simply providing a static goal configuration, which would be very difficult to solve for the RL methods if only reward is computed at the end-state. In fact, when the goal pattern is relatively simple, we can compute particle correspondences with the goal state easily if we use a straight line initial actions, and our methods can use loss information of the last state alone and produce meaningful trajectories. [Here](https://s3.us-west-2.amazonaws.com/secure.notion-static.com/4075ef8d-828a-4758-896b-15789f8524cf/output.mp4?X-Amz-Algorithm=AWS4-HMAC-SHA256&X-Amz-Content-Sha256=UNSIGNED-PAYLOAD&X-Amz-Credential=AKIAT73L2G45EIPT3X45%2F20221117%2Fus-west-2%2Fs3%2Faws4_request&X-Amz-Date=20221117T013226Z&X-Amz-Expires=86400&X-Amz-Signature=8ab2c0083ab9ddd55bf084d7eb0a478d2cf9ff24ddc701d4d5cc6f0acd336d32&X-Amz-SignedHeaders=host&response-content-disposition=filename%3D%22output.mp4%22&x-id=GetObject) is a simple example of optimized trajectory from DP using only a static goal state, where the goal pattern is a sine wave. However, with a more complex goal pattern, it's very difficult to obtain a reasonable particle level correspondence. This is also a valuable future research direction and we included it in the discussion section in our paper.
> > >
> > > ### 5. Code release
> > > > Reproducibility: considering the system is complex, without the code, it is hard for one to judge the reproducibility.
> > >
> > > We have released our code anonymously here!: https://github.com/FluidLabICLR2023/FluidLab We will release another version of code with better cleaning up and more detailed instructions.
> > >
> > > We hope our response could address your major concerns. We would like to thank you again for the critical and helpful feedbacks and suggestions! They have significantly strengthened the quality of our work.

---

### Author Response · Authors · 2022-11-17
**General Response**

We thank all the reviewers for their detailed feedback and useful suggestions! We are very encouraged that the reviewers found our work presents interesting, challenging, complex and diverse fluid manipulation tasks (Frwu, 3F7A, KJH8, fmMN), introduces novel optimization techniques (KJH8), our proposed physics engine is a significant advance and valuable (KJH8, fmMN), and our paper is of decent quality, clearly written and well-organized (Frwu, KJH8, fmMN) with strong experimental results (fmMN).

We have tried our best to address all the concerns of the reviewers in the individual responses. We have also updated our manuscript and [project website](https://sites.google.com/view/fluidlab) with requested experiments and discussions. We summarize the major changes below:
- We updated our manuscript and project website with additional **real robot execution** results. (Appendix E and **[visual results here](https://sites.google.com/view/fluidlab/real-world-results)**) [Frwu, KJH8]
- We added detailed comparison with **other differentiable simulation environments** including SPNets, DiffSim, Disect, DiffTaichi, JAX-Fluids, and PhiFlow. (Table 1 and Appendix G) [Frwu, 3F7A]
- We included additional evaluations of **PODS**, a method that combines RL and differentiable simulation. (Section 5.2) [Frwu]
- We added a subsection with an analysis of our experimental results and discussions on potential **future research directions** (the last paragraph in Section 5.2) [Frwu]
- We added a section describing the **reasoning and motivations** behind our task selection choices, and also revised the ordering of our tasks. (Appendix H) [3F7A]
- We added a series of **validation experiments** to validate the accuracy and reliability of our simulation engine. (Appendix F and **[visual results here](https://sites.google.com/view/fluidlab/fluidengines-validation-exps)**) [Frwu, KJH8]
- We included additional **details of simulations** of our task settings (Appendix D.1) [fmMN]
- We released our **code** as an anonymous github repo here: https://github.com/FluidLabICLR2023/FluidLab. [Frwu, KJH8, fmMN]

We hope our responses could address all reviewers' concerns convincingly. We would like to express our gratitude towards all the reviewers again for their detailed and constructive feedbacks!

---

> ### Author Response · Authors · 2023-02-09
> **Code Release**
>
> The anonymous github repo is now transferred to the author with a new link: https://github.com/zhouxian/FluidLab

---

### Decision · Program_Chairs · 2023-01-20

**Decision:**

Accept: notable-top-25%

**Justification For Why Not Higher Score:**

While this is a strong experimental paper, no new methodology has been introduced.
Could be an oral, but I do not immediately see a reason for it.

**Justification For Why Not Lower Score:**

Could be also a poster

**Metareview: Summary, Strengths And Weaknesses:**

This paper presents a new simulation environment for different manipulation tasks involving fluids. It had received 4 expert reviews with a relatively high rating average of 7, but an initially high spread, with two yet unconvinced reviewers. Their concerns were on missing benchmarking of existing methods and on positioning.

The authors provided answers and could address most of the comments and issues raised by the reviewers.

A consensus quickly emerged on the quality of the work: the introduction of a useful new software and benchmark, requiring more interactions than existing work. The AC concurs and recommends acceptance.

**Note From Pc:**

if the above contains the word "oral" or "spotlight" please see: "oral" presentation means -> notable-top-5% and "spotlight" means -> notable-top-25%. As stated in our emails, we are disassociating presentation type from AC recommendations

**Summary Of Ac-Reviewer Meeting:**

No meeting was held.